# A sodium-ion-conducted asymmetric electrolyzer to lower the operation voltage for direct seawater electrolysis

Hao Shi[1], Tanyuan Wang [1]✉, Jianyun Liu[1], Weiwei Chen[2], Shenzhou Li[1], Jiashun Liang[1], Shuxia Liu[1], Xuan Liu[1], Zhao Cai[3], Chao Wang[4], Dong Su [2], Yunhui Huang [1], Lior Elbaz [5] & Qing Li [1]✉

Hydrogen produced from neutral seawater electrolysis faces many challenges including high energy consumption, the corrosion/side reactions caused by $Cl^-$, and the blockage of active sites by $Ca^{2+}/Mg^{2+}$ precipitates. Herein, we design a pH-asymmetric electrolyzer with a $Na^+$ exchange membrane for direct seawater electrolysis, which can simultaneously prevent $Cl^-$ corrosion and $Ca^{2+}/Mg^{2+}$ precipitation and harvest the chemical potentials between the different electrolytes to reduce the required voltage. In-situ Raman spectroscopy and density functional theory calculations reveal that water dissociation can be promoted with a catalyst based on atomically dispersed Pt anchored to Ni-Fe-P nanowires with a reduced energy barrier (by 0.26 eV), thus accelerating the hydrogen evolution kinetics in seawater. Consequently, the asymmetric electrolyzer exhibits current densities of 10 mA cm$^{-2}$ and 100 mA cm$^{-2}$ at voltages of 1.31 V and 1.46 V, respectively. It can also reach 400 mA cm$^{-2}$ at a low voltage of 1.66 V at 80 °C, corresponding to the electricity cost of US\$1.36 per kg of $H_2$ (\$0.031/kW h for the electricity bill), lower than the United States Department of Energy 2025 target (US\$1.4 per kg of $H_2$).

The growing need for sustainable energy sources in order to reduce greenhouse gas (GHGs) emissions, and increase energy independence and resilience requires a sustainable and economically viable solution for large energy storage. Hydrogen is considered as the most promising alternative to fossil fuels, owing to its high energy density (142 MJ Kg$^{-1}$) and emissions-free use[1,2]. Water electrolysis is an efficient and sustainable route to produce high-purity $H_2$, which can make full use of sustainable, yet intermittent energy sources supplied by wind, solar, etc.[3]. However, the large-scale application of electrolyzers using high-purity deionized water may be limited[4]. Seawater is the most significant natural resource which accounts for 96.5% of the total water on earth, but it is still underutilized for water electrolysis[5,6]. Moreover,

direct seawater electrolysis can take full advantage of the seaside wind energy, with a potential of producing more than 13,500 terawatt hours per year in the United States alone, approximately 3 times its annual electricity consumption[7,8], which makes seawater electrolysis a very appealing approach for green $H_2$ production.

However, natural seawater is a complex neutral electrolyte (pH ~ 8), containing multiple ions, among which the main constituents are $Cl^-$ and $Na^+$[5]. The major issues associated with direct seawater electrolysis are: (1) the undesired anodic chloride oxidation reactions (ClOR), i.e., the chlorine evolution ($2Cl^- \rightarrow Cl_2 + 2e^-$) and the formation of hypochlorite ($Cl^- + 2OH^- \rightarrow ClO^- + H_2O + 2e^-$), competing with the oxygen evolution reaction (OER)[9]; (2) the high energy cost caused by the

[1]State Key Laboratory of Material Processing and Die & Mould Technology, School of Materials Science and Engineering, Huazhong University of Science and Technology, 430074 Wuhan, Hubei, China. [2]Beijing National Laboratory for Condensed Matter Physics, Institute of Physics, Chinese Academy of Sciences, 100190 Beijing, China. [3]Faculty of Materials Science and Chemistry, China University of Geosciences (Wuhan), 430074 Wuhan, Hubei, China. [4]School of Materials Science and Engineering, Tongji University, 201804 Shanghai, China. [5]Department of Chemistry and the Institute of Nanotechnology and Advanced Materials, Bar-Ilan University, 5290002 Ramat-Gan, Israel. ✉e-mail: wangty@hust.edu.cn; qing_li@hust.edu.cn

sluggish reaction kinetics for hydrogen evolution reaction (HER) and OER in the neutral medium without a buffer[10]; and (3) other ions such as $Ca^{2+}$ and $Mg^{2+}$ that may precipitate and block the catalytic active sites[11]. These issues pose great challenges to the design of electrolyzers and catalysts for efficient direct seawater electrolysis. Most of the research has been focused on alkaline seawater electrolysis, for which alkali is added to the seawater to suppress the evolution of chlorine[12–14]. However, chloride still remain in these systems to some extent, and they generally suffer from inferior activity compared with alkaline electrolyzers working with deionized water[15]. More recently, an asymmetric electrolyte feed method with an anion exchange membrane as the separator has been reported for seawater electrolysis[9], but Cl⁻ still seem to cross the membrane to the anode under the electric field, which not only lead to undesirable ClOR at the anode but also destroy the membrane. Moreover, the asymmetric feed electrolyzer demonstrates an inferior performance to that of symmetric KOH feed due to the sluggish kinetics of HER in natural seawater without buffer media. Therefore, it is highly desirable to develop an electrolysis system with efficient and robust catalysts that will inhibit ClOR and realize direct seawater electrolysis under low voltage[16].

Herein, we present our newly developed pH-asymmetric feed electrolyzer using a Na⁺ exchange membrane as a separator for efficient and energy-saving direct seawater electrolysis for the first time. The Na⁺ exchange membrane could prevent the transport of Cl⁻ ions in the catholyte (NaCl solution or natural seawater) to the anode (NaOH), thus avoiding the competing ClOR. The problem of $Ca^{2+}$ and $Mg^{2+}$ precipitates will be alleviated for the HER in the near-neutral seawater (pH <9.5) and flow electrolyte. The energy cost of the electrolyzer can also be significantly reduced due to the pH-asymmetric framework as well as the accelerated reaction kinetics benefited from rationally designed amorphous $Ni_{6.6}Fe_{0.4}P_3$ NWs supported Pt single atoms (denoted as $Pt_{SA}-Ni_{6.6}Fe_{0.4}P_3$) and amorphous $Ni_5Fe_2P_3$ NWs as efficient anode and cathode catalysts, respectively. As a result, the asymmetric electrolyzer displays the current densities of 10 mA cm⁻² and 100 mA cm⁻² at voltages of 1.31 V and 1.46 V, respectively, outperforming the state-of-the-art electrolyzers for seawater electrolysis[15,17–27]. In addition, it can reach a current density of 400 mA cm⁻² at a low voltage of 1.66 V at 80 °C (iR-compensation-free), corresponding to an electricity cost of US\$1.36 per kg of $H_2$ (3.96 kWh per m³ $H_2$, \$0.031/kW h for the electricity bill), which is lower than the U.S. Department of Energy (DOE) 2025 target of US\$1.4 per kg of $H_2$[28]. The accelerated water dissociation process as well as the super-hydrophilic surface for $Pt_{SA}-Ni_{6.6}Fe_{0.4}P_3$ are suggested to account for the enhanced catalytic performance.

## Results

### Electrolyzer design and electrocatalyst characterizations

The scheme of the electrolyzer with an asymmetric electrolyte feed developed in this work is shown in Fig. 1a. Bipolar chambers are separated by treated Na⁺ exchange membrane (details in the Methods section), which prevents the Cl⁻ transportation through the membrane to the anode, thus avoiding the undesired ClOR[29]. Both the anode and cathode catalysts are supported on porous Ni electrodes which also serves as the flow channels at the same time. In the asymmetric electrolyzer, NaCl solution, or natural seawater is circulated in the cathode chamber, while NaOH solution is circulated in the anode chamber. Since both half reactions in water splitting are pH-sensitive redox reactions, as manifested in the Pourbaix diagram of water (Fig. 1b), the chemical potential difference between electrolytes of different pH in cathode and anode can be used to reduce the required voltage (from 1.23 V to 0.816 V) for direct seawater electrolysis[30]. The dependence of the electrolyzer half reactions and corresponding electrode potentials as function of the electrolyte pH are given in Supplementary Note 1. The issue with $Mg^{2+}$ and $Ca^{2+}$ precipitates in seawater electrolyzer could be alleviated in the asymmetric electrolyzer with Na⁺ ions exchange membrane, since these hydroxide precipitates are usually formed at pH >9.5 and greatly influenced by the electrolyte flow rate[31].

In order to lower the energy consumption in direct seawater electrolysis, efficient catalysts should be rationally designed. Herein, we display a simple method to synthesize amorphous Ni-Fe-P NWs and single-atom Pt loaded amorphous Ni-Fe-P NWs as OER and HER catalysts, respectively. In brief, we used an electrodeposition method to form hierarchically porous Ni-Fe alloy on Ni foam (Supplementary Fig. 1), followed by a solvothermal method[13] where the Ni-Fe alloy reacted with red phosphorus in diethylene glycol to form Ni-Fe-P NWs. $Ni_xFe_{1-x}P_3$ with varying ratios of Ni to Fe can be easily synthesized by adjusting the ratio of $Ni^{2+}$ and $Fe^{2+}$ in the electrolyte during electrodeposition (for details, see Methods section). The actual elemental compositions of these catalysts were investigated using X-ray Fluorescence Spectrometry (XRF) and the results are summarized in

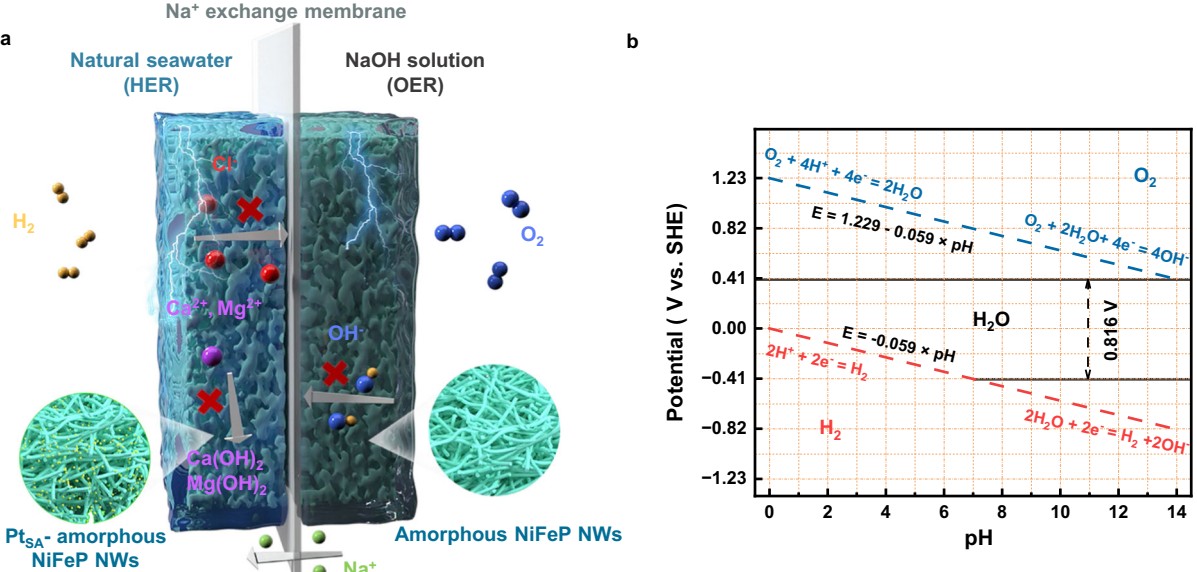

**Fig. 1 | Design strategies of pH-asymmetric electrolyzer. a** Scheme for the asymmetric electrolyzer with sodium ions exchange membrane. **b** The Pourbaix diagram of water.

Supplementary Table 1. Scanning electron microscopy (SEM; Supplementary Fig. 2) images show that interwoven $Ni_xFe_{1-x}P_3$ NWs were in situ formed inside the Ni foam after phosphatizing. X-ray diffraction (XRD) patterns show that only the peaks of Ni foam can be detected, which indicate the amorphous nature of $Ni_xFe_{1-x}P_3$ NWs (Supplementary Fig. 3). $Pt_{SA}$-$Ni_{6.6}Fe_{0.4}P_3$ catalyst was obtained by electroreduction process with cyclic voltammetry in 1 M NaOH containing ultralow Pt salts concentration, and single atoms of Pt (1.35 wt%, measured using inductively coupled plasma-mass spectrometry) were anchored on the defect-rich amorphous $Ni_{6.6}Fe_{0.4}P_3$ NWs due to its disordered long-range atomic arrangement[32]. The XRD pattern of $Pt_{SA}$-$Ni_{6.6}Fe_{0.4}P_3$ on Ni foam is shown in Fig. 2a, in which no Pt characteristic peaks were detected, implying the absence of crystalline Pt. In addition, the XRD pattern (Supplementary Fig. 4) of the powder that was scraped from the surface of the $Pt_{SA}$-$Ni_{6.6}Fe_{0.4}P_3$ displays no obvious peaks, which further proves the amorphous feature of $Pt_{SA}$-$Ni_{6.6}Fe_{0.4}P_3$. SEM and transmission electron microscopy (TEM) images also support that $Pt_{SA}$-$Ni_{6.6}Fe_{0.4}P_3$ form an interwoven NWs framework with a NW diameter of about 60 nm (Fig. 2b, c). Moreover, no lattice fringes can be observed on the $Pt_{SA}$-$Ni_{6.6}Fe_{0.4}P_3$ NW (Fig. 2d). The fast Fourier transform (FFT) image (inset of Fig. 2d) of $Pt_{SA}$-$Ni_{6.6}Fe_{0.4}P_3$ also displays a diffuse center spot, indicating its maintaining of amorphous phase, which is consistent with the result of XRD patterns. In addition, elemental mappings of $Pt_{SA}$-$Ni_{6.6}Fe_{0.4}P_3$ confirm the presences and uniform distributions of Pt, Ni, Fe and P on the NW (Fig. 2e). The high-angle annular darkfield STEM (HAADF-STEM, Supplementary Fig. 5) was used to investigate the distribution of Pt over $Ni_{6.6}Fe_{0.4}P_3$. The bright spots appear on the amorphous structure, corresponding to heavy constituent atom species, which clearly confirms the immobilization of atomically dispersed Pt atoms.

X-ray photoelectron spectroscopy (XPS) was used to study the surface element states of the catalyst. As shown in Fig. 3a, the peaks of Pt 4$f$ for $Pt_{SA}$-$Ni_{6.6}Fe_{0.4}P_3$ shift positively (-0.3 eV) compared to the

$Pt^0$ peaks for Pt foil, indicating the electrons are transformed from Pt to the $Ni_{6.6}Fe_{0.4}P_3$ support. Notably, the peak at 68.3 eV is attributed to Ni 3$p$, which can also be detected for $Ni_{6.6}Fe_{0.4}P_3$ support[33]. The XPS spectra of Ni and P in the $Pt_{SA}$-$Ni_{6.6}Fe_{0.4}P_3$ before and after the addition of Pt are shown in Supplementary Fig. 6, in which the peaks of P 2$p$ show a negative shift, further demonstrating the electronic interaction between Pt and the support. X-ray absorption near-edge spectroscopy (XANES) of $Pt_{SA}$-$Ni_{6.6}Fe_{0.4}P_3$, standard Pt foil, and $PtO_2$ are shown in Fig. 3b. the white-line intensity has been reported to correspond to the transfer of the Pt $2p_{3/2}$ core-electron to 5$d$ states, and thus is used as an indicator for Pt 5$d$-band occupancy[34]. The slightly higher white-line intensity of $Pt_{SA}$-$Ni_{6.6}Fe_{0.4}P_3$ than that of Pt foil indicates the more unoccupied 5$d$ orbitals for single-atom Pt, which further certifies the charge loss of the single-atom Pt after coordinating with the supports, in line with the XPS analysis in Fig. 3a. To capture the atomic coordination information of Pt, the extend X-ray absorption fine structure (EXAFS) analysis was carried out on the $Pt_{SA}$-$Ni_{6.6}Fe_{0.4}P_3$ catalyst. Figure 3c shows the $k^3$-weighted Fourier transform (FT) curves at R space of Pt L3-edge EXAFS spectra for $Pt_{SA}$-$Ni_{6.6}Fe_{0.4}P_3$ in comparison with the references of $PtO_2$ and Pt foil. It shows that the peak of typical Pt-Pt bond for Pt foil at about 2.7 Å is absent in the $Pt_{SA}$-$Ni_{6.6}Fe_{0.4}P_3$, which strengthens the postulate that the Pt is atomically dispersed and does not form nanoparticles or small clusters. The peak at 2.1 Å for $Pt_{SA}$-$Ni_{6.6}Fe_{0.4}P_3$ is associated with the Pt-Ni/Fe bonds, in which the Pt-Ni and Pt-Fe bonds are difficult to distinguish due to the similar bond lengths[34,35]. In addition, the wavelet transform analysis (Fig. 3d–f) was carried out to further prove the single-atom Pt dispersion on $Ni_{6.6}Fe_{0.4}P_3$[36,37]. The intensity maximum at 11 Å$^{-1}$ which corresponds to Pt-Pt is absent for $Pt_{SA}$-$Ni_{6.6}Fe_{0.4}P_3$, which also confirms that the Pt atoms are atomically dispersed on the catalysts. A main Pt-Ni coordination is taken to simplify the modeling of this system in EXAFS fitting due to the relatively large Ni:Fe ratio (33:2). The first-shell EXAFs fitting of

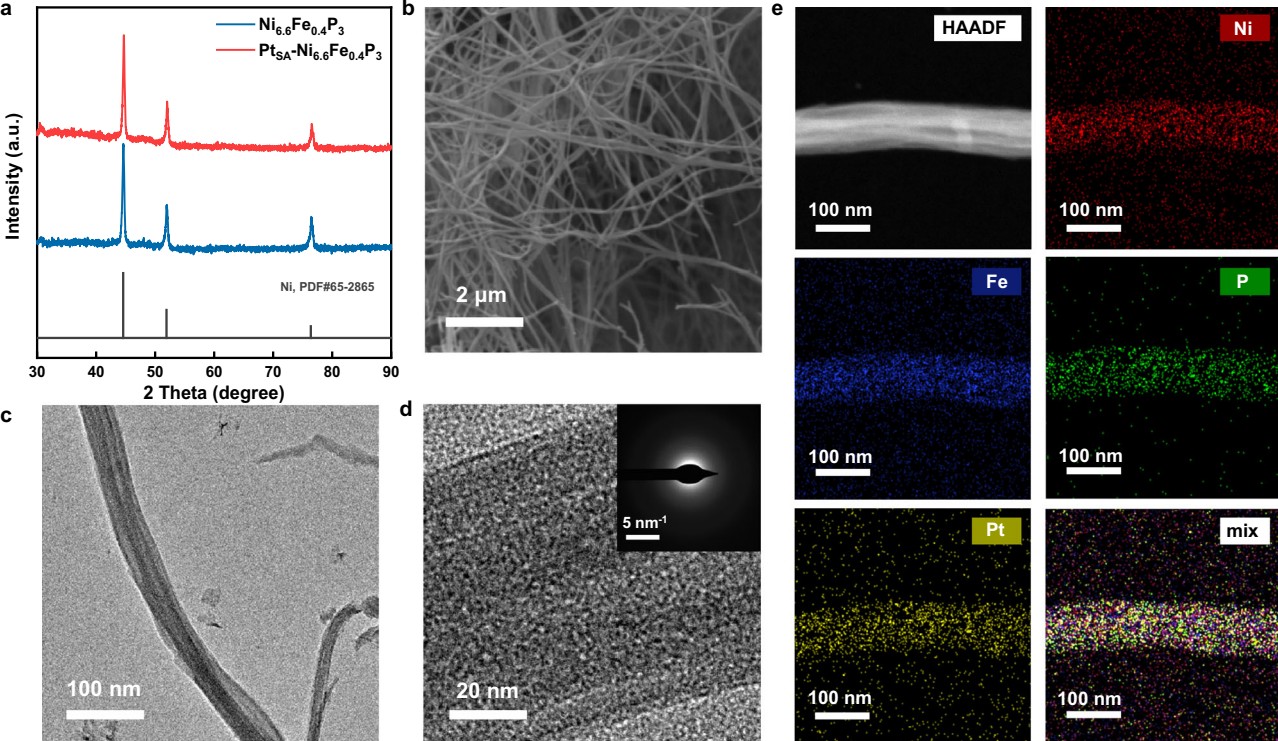

**Fig. 2 | The morphology and structure characterization of the fabricated $Pt_{SA}$-$Ni_{6.6}Fe_{0.4}P_3$ catalyst. a** XRD patterns of $Pt_{SA}$-$Ni_{6.6}Fe_{0.4}P_3$ and $Ni_{6.6}Fe_{0.4}P_3$. **b** SEM image of $Pt_{SA}$-$Ni_{6.6}Fe_{0.4}P_3$ NWs. **c** TEM image of $Pt_{SA}$-$Ni_{6.6}Fe_{0.4}P_3$ NWs. **d** HRTEM image of $Pt_{SA}$-$Ni_{6.6}Fe_{0.4}P_3$ (inset shows the corresponding FFT). **e** HAADF-STEM image and the corresponding EDS elemental mapping analysis of $Pt_{SA}$-$Ni_{6.6}Fe_{0.4}P_3$.

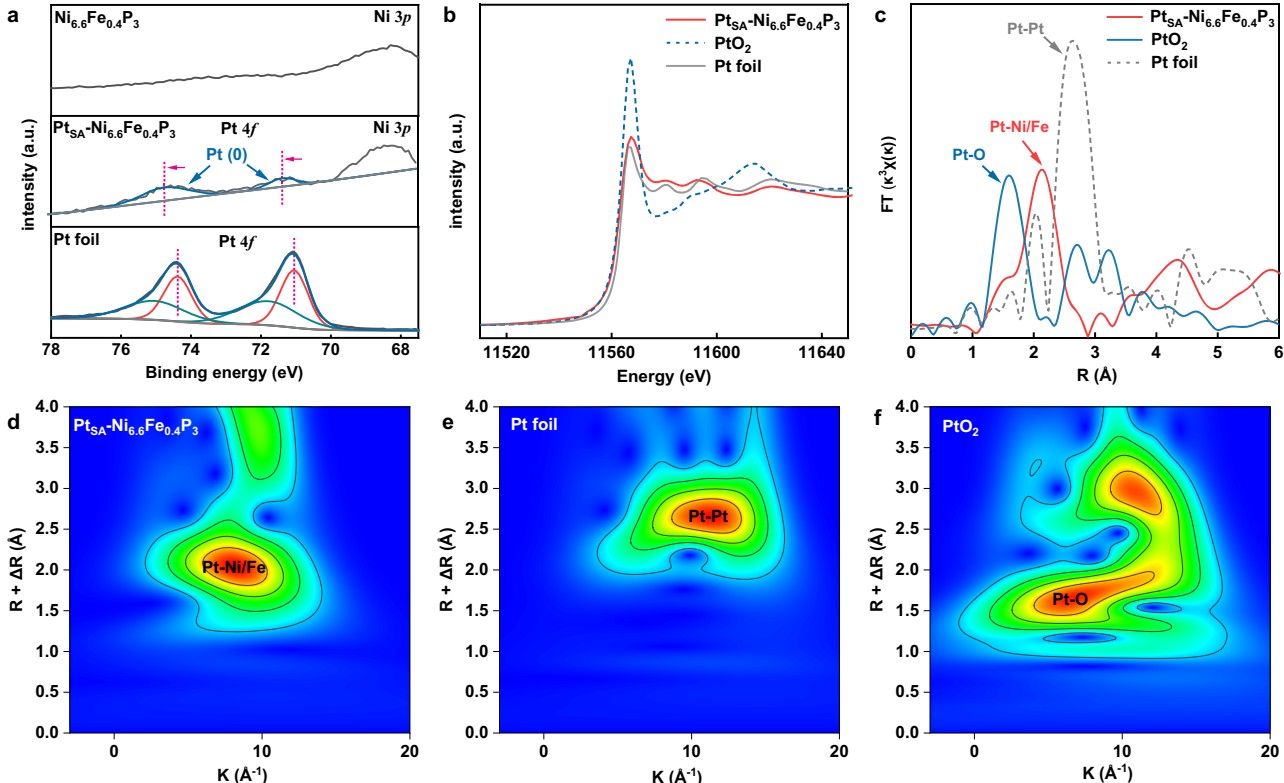

**Fig. 3 | Fine structure characterizations of Pt$_{SA}$-Ni$_{6.6}$Fe$_{0.4}$P$_3$ catalyst. a** XPS spectra of Pt 4$f$ for Pt$_{SA}$-Ni$_{6.6}$Fe$_{0.4}$P$_3$ and Pt foil. **b** Pt L3-edge XANES spectra of Pt$_{SA}$-Ni$_{6.6}$Fe$_{0.4}$P$_3$, PtO$_2$ and Pt foil. **c** Corresponding Pt L3 $k^3$-weighted FT-EXAFS curves of Fig. 3b. **d**–**f** EXAFS wavelet transform plots of Pt$_{SA}$-Ni$_{6.6}$Fe$_{0.4}$P$_3$, PtO$_2$, and Pt foil.

Pt$_{SA}$-Ni$_{6.6}$Fe$_{0.4}$P$_3$ gives a coordination number of 1.0 for Pt-O contribution and 3.3 for Pt-Ni contribution (Supplementary Fig. 7 and Supplementary Table 2), suggesting that the Pt atoms are mainly anchored at the hollow sites between Ni atoms.

**Electrocatalytic performance for seawater electrolysis**

The HER performance was evaluated first with the as-prepared catalysts in 4 M NaCl solution. For comparison, the Ni-Fe alloy supports Pt single atoms (denoted as Pt$_{SA}$-Ni-Fe, Methods), Pt/C (20 wt%), Ni foam and Ni$_{6.6}$Fe$_{0.4}$P$_3$ were also studied under the same conditions (Fig. 4a and Supplementary Fig. 8). The Pt$_{SA}$-Ni$_{6.6}$Fe$_{0.4}$P$_3$ exhibits the highest HER activity among all of the above-mentioned catalysts, and requires an overpotential of 408 mV for a HER current density of 100 mA cm$^{-2}$. The overpotentials required to achieve the same current density with Pt$_{SA}$-Ni-Fe (551 mV) and Pt/C (673 mV) are much higher, which hints on the contribution of the amorphous Ni$_{6.6}$Fe$_{0.4}$P$_3$ substrate to the HER electrocatalytic activity of the atomically dispersed Pt sites. In addition, the Pt$_{SA}$-Ni$_{6.6}$Fe$_{0.4}$P$_3$ exhibits a smaller Tafel slope of 131 mV dec$^{-1}$ than that of Pt$_{SA}$-Ni-Fe and commercial Pt/C catalysts as shown in Supplementary Fig. 9, which indicates the faster HER kinetic of Pt$_{SA}$-Ni$_{6.6}$Fe$_{0.4}$P$_3$[38]. The HER activity of Pt$_{SA}$-Ni$_x$Fe$_{1-x}$P$_3$ catalysts with different Ni:Fe ratios were also studied here, and the results are presented in Supplementary Fig. 10. Among these catalysts, the Pt$_{SA}$-Ni$_{6.6}$Fe$_{0.4}$P$_3$ with the Ni:Fe ratio of 6.6:0.4 shows the best HER performance. In addition, the electrochemical active surface area (ECSA) was calculated for different supports by measuring the double-layer capacitance ($C_{dl}$) from their CVs (Supplementary Fig. 11). The $C_{dl}$ of Ni$_{6.6}$Fe$_{0.4}$P$_3$ is 17.30 mF cm$^{-2}$, relatively high, owing to its NW morphology, and is 2.4 times higher than that of Ni-Fe alloy (7.09 mF cm$^{-2}$), demonstrating the highly improved ECSA and the possibly increased number of active sites after P doping. Interestingly, the Ni$_x$Fe$_{1-x}$P$_3$ catalysts with different Ni:Fe ratios exhibit similar ECSA, indicating that the intrinsic activity of the active sites for the catalysts are different. The HER performance of

the different catalysts measured in 4 M NaCl and normalized by ESCA are presented in Supplementary Fig. 12. There, the ECSA-normalized performance of the Pt$_{SA}$-Ni$_{6.6}$Fe$_{0.4}$P$_3$ is the highest of all in terms of current density, indicating its high intrinsic HER activity. Furthermore, the mass activity of Pt$_{SA}$-Ni$_{6.6}$Fe$_{0.4}$P$_3$ measured at an overpotential of 400 mV and normalized to the Pt loading is 2.17 A mg$^{-1}$, about 12 times greater than that of commercial Pt/C (0.18 A mg$^{-1}$) (Supplementary Fig. 13), emphasizing that the atomically dispersed Pt supported on Ni$_{6.6}$Fe$_{0.4}$P$_3$ could promote HER in neutral media better than commercial Pt/C. The HER activity of Pt$_{SA}$-Ni$_{6.6}$Fe$_{0.4}$P$_3$ in 1 M NaCl and natural seawater are also investigated (Supplementary Fig. 14). After 90% iR-compensation, the Pt$_{SA}$-Ni$_{6.6}$Fe$_{0.4}$P$_3$ catalyst reveals the similar HER activity in 4 M NaCl, 1 M NaCl and natural seawater. To better illustrate the stability of the Pt$_{SA}$-Ni$_{6.6}$Fe$_{0.4}$P$_3$ catalyst in natural seawater, the HER stability test in natural seawater for Pt$_{SA}$-Ni$_{6.6}$Fe$_{0.4}$P$_3$ with a large container is investigated (Supplementary Fig. 15). There is almost no performance degradation with the stirring of the solution at 100 mA cm$^{-2}$ for 18 h, indicating the durability of Pt$_{SA}$-Ni$_{6.6}$Fe$_{0.4}$P$_3$. Moreover, Pt$_{SA}$-Ni$_{6.6}$Fe$_{0.4}$P$_3$ demonstrates excellent HER performance in 1 M NaOH and 1 M PBS (phosphate-buffered saline, pH = 7, Supplementary Figs. 16 and 17). The Pt$_{SA}$-Ni$_{6.6}$Fe$_{0.4}$P$_3$ shows low overpotentials of 89 mV and 240 mV required to reach 100 mA cm$^{-2}$ in 1 M NaOH and 1 M PBS, respectively, which is one of the best reported HER catalysts[34,39].

Since the HER performance at high current densities is greatly influenced by the wettability of the electrode surface[40], contact-angle measurements were performed with Pt$_{SA}$-Ni$_{6.6}$Fe$_{0.4}$P$_3$ electrode. As shown in Fig. 4b, Pt$_{SA}$-Ni$_{6.6}$Fe$_{0.4}$P$_3$ is superhydrophilic, as the droplets spread immediately once in contact with the surface (details in Supplementary Fig. 18). The underwater gas-bubble contact-angles were measured as 132° and 126° for Pt$_{SA}$-Ni$_{6.6}$Fe$_{0.4}$P$_3$ and Ni foam, demonstrating the lower adhesive force to the bubbles for Pt$_{SA}$-Ni$_{6.6}$Fe$_{0.4}$P$_3$[41]. The smaller and denser bubbles during HER (Supplementary Fig. 19) for

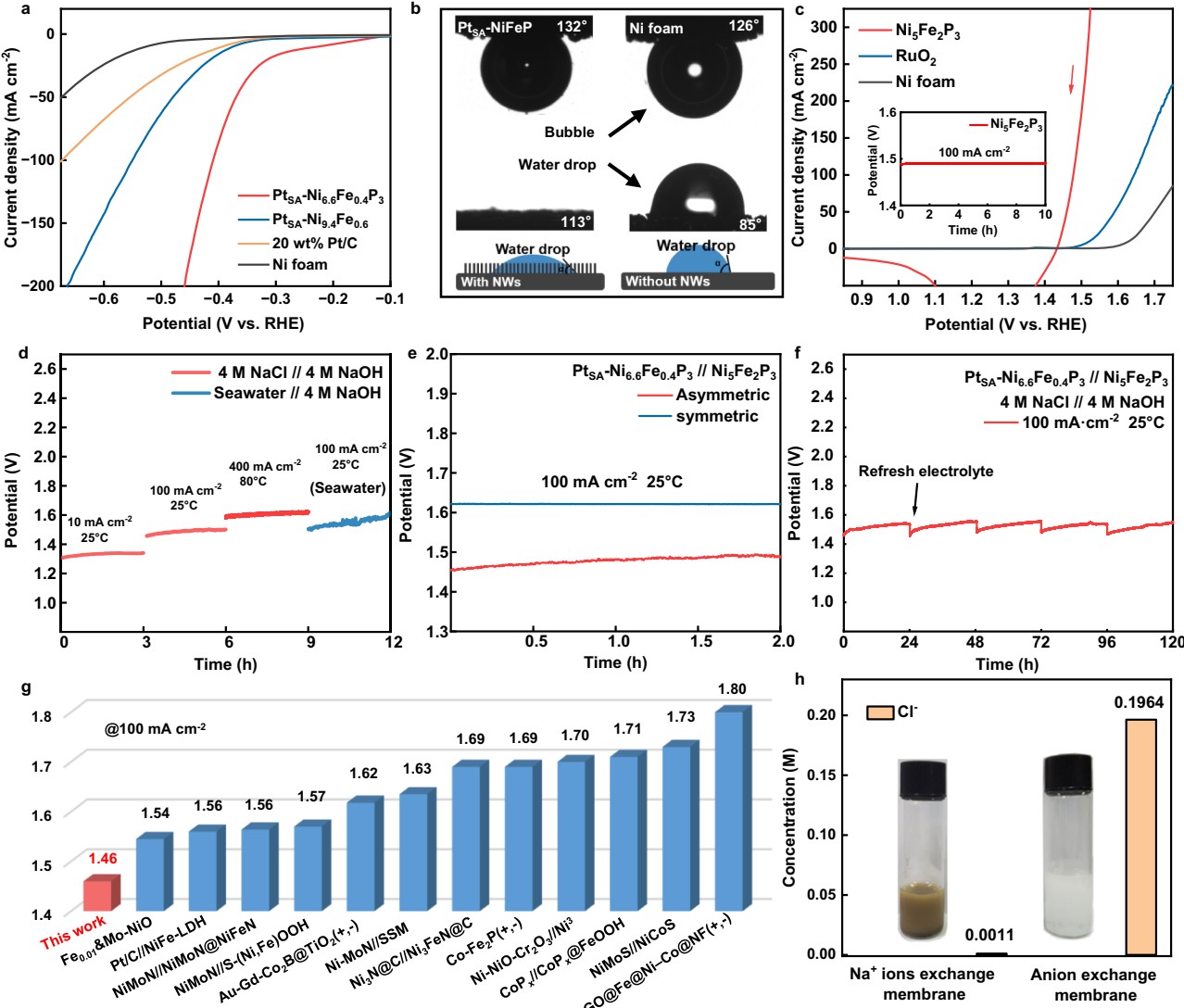

**Fig. 4 | Electrochemical performance. a** HER polarization curves of Pt$_{SA}$-Ni$_{6.6}$Fe$_{0.4}$P$_3$, Pt$_{SA}$-Ni-Fe, Pt/C and Ni foam in 4 M NaCl with 90% iR-compensation. (resistance value: $1.1 \pm 0.1$ Ω) **b** Air-bubble contact angles under water (top), staticwater-droplet contact angles (middle) and the structure Schematic illustration (bottom, Wenzel model) for Pt$_{SA}$-Ni$_{6.6}$Fe$_{0.4}$P$_3$ and Ni foam. **c** OER polarization curves of Ni$_5$Fe$_2$P$_3$, RuO$_2$ and Ni foam in NaOH and chronopotentiometry curve (inset) with 90% iR-compensation. (resistance value: $1.3 \pm 0.1$ Ω) **d** Chronopotentiometry curves of asymmetric electrolyzers at different conditions

(catalyst size: $0.5 \times 1$ cm$^2$, Pt loading: 40 μg). **e** Chronopotentiometry curve of asymmetric and symmetric electrolyzers at 100 mA cm$^{-2}$. **f** Long-term stability test of the asymmetric electrolyzer at constant 100 mA cm$^{-2}$. **g** Operation voltage of the state-of-the-art seawater electrolyzers for overall seawater electrolysis at 100 mA cm$^{-2}$. **h** Concentration of Cl$^-$ ions of anode electrolytes for asymmetric electrolyzers with Na$^+$/anion exchange membrane and the photographs after adding AgNO$_3$ solution (inset).

Pt$_{SA}$-Ni$_{6.6}$Fe$_{0.4}$P$_3$ also confirms this observation. The superhydrophilic Pt$_{SA}$-Ni$_{6.6}$Fe$_{0.4}$P$_3$ with lower adhesive force to the bubbles results in fast bubble release from the electrode surface, thus exposing more active sites and improving its HER performance in NaCl solution[41].

The Ni$_x$Fe$_{1-x}$P$_3$ support also exhibits excellent OER activity in 1 M NaOH. The OER polarization curve was collected by the negative sweeping to exclude the interference of Ni$^{2+}$ oxidation current with the actual OER current (Supplementary Fig. 20)[42]. As shown in Fig. 4c and Supplementary Fig. 21, the Ni$_5$Fe$_2$P$_3$ shows the highest OER performance among all the catalysts with a quite low overpotential of 248 mV to reach the current density of 100 mA cm$^{-2}$. More importantly, the catalyst shows no degradation over a period of 10 h continuous operation (inset in Fig. 4c), confirming its excellent stability in half-cell. Tafel plots constructed from this data show that the Ni$_5$Fe$_2$P$_3$ has a relatively smaller slope of 30 mV dec$^{-1}$ in comparison with that of the RuO$_2$ (63 mV dec$^{-1}$) and Ni foam (81 mV dec$^{-1}$), verifying its rapid OER catalytic kinetics (Supplementary Fig. 22)[43].

The rotating ring-disk electrode technique was used to quantitatively detect the local pH on the Pt$_{SA}$-Ni$_{6.6}$Fe$_{0.4}$P$_3$ for HER in NaCl solution (Supplementary Note 2 and Supplementary Figs. 23 and 24)[44]. We find that the pH value on the Pt$_{SA}$-Ni$_{6.6}$Fe$_{0.4}$P$_3$ increases when HER occurs but it is still far lower than the anode pH (14.4), indicating that the chemical potential due to the pH difference can be leveraged to increase the efficiency of the process. The developed catalysts were then applied for the designed asymmetric electrolyzer with Pt$_{SA}$-Ni$_{6.6}$Fe$_{0.4}$P$_3$ and Ni$_5$Fe$_2$P$_3$ as cathode and anode catalysts, respectively. The cathode chamber was circuited with 4 M NaCl solution and anode chamber was circuited with 4 M NaOH solution to maintain the balance of the concentration for Na$^+$ between two half cells. The electrolyzer assembly is presented in Supplementary Fig. 25. Remarkably, the asymmetric electrolyzer reaches the current density of 10 mA cm$^{-2}$ at a low voltage of 1.31 V, and a current density of 100 mA cm$^{-2}$ at 1.46 V and 1.50 V for 4 M NaCl solution and natural seawater as cathode electrolyte, respectively (Fig. 4d), which outperform the state-of-the-art

electrolyzers operating with seawater. Meanwhile, the symmetric electrolyzer with the same catalysts requires a voltage of 1.62 V to reach a current density of 100 mA cm$^{-2}$ with 4 M NaOH as the electrolyte (Fig. 4e), indicating that the asymmetric electrolyzer with the above catalysts could significantly reduce the energy cost of water electrolysis. The long-term stability performance of the electrolyzer are further shown in Fig. 4f and Supplementary Fig. 26. Slight voltage increases were observed, which may due to increase pH of the NaCl solution during long-term test, in which the proton will be consumed if the solution is cycled. The pH of anode and cathode electrolyte of the asymmetric electrolyzer over time at 100 mA cm$^{-2}$ was also monitored over 12 h when the electrolytes were cycled. As shown in Supplementary Fig. 27, the pH of cathode electrolyte increases from 6.9 to 8.8, and the pH of anode electrolyte slightly drops from 14.4 to 14.1, which may be responsible for the performance degradation of the asymmetric electrolyzer for water electrolysis. However, the pH of the output cathode electrolyte (Supplementary Fig. 28) could be maintained at about 8.5 when operating at 100 mA cm$^{-2}$ in one-way flow scheme, and the electrolyte with stable pH can exhibit stable voltage response. After 120 h water splitting test, the voltage required to reach a current density of 100 mA cm$^{-2}$ shows only very little increase after refreshing the solution, verifying the superior durability of the system. When using the asymmetric electrolyzer working under industrial operating conditions of 400 mA cm$^{-2}$ and 80 °C for water splitting with one-way flow scheme, the operating voltage is only 1.66 V without iR-compensation (Supplementary Fig. 29), which is also significantly lower than the symmetric electrolyzer (1.83 V, Supplementary Fig. 30). The stable voltage proves the stability for the asymmetric electrolyzer with one-way flowing feed scheme to stabilize the pH on the cathode. The electricity cost of the asymmetric electrolyzer is reduced to 3.96 kWh per m$^3$ H$_2$, corresponding to US\$1.36 per kg of H$_2$, which is lower than the U.S. DOE 2025 target of US\$1.4 per kg of H$_2$ and may be further reduced through engineering system architecture. The total hydrogen levelized cost is estimated at US\$1.96 per kg of H$_2$, slightly lower than U.S. DOE 2025 target of US\$2 per kg of H$_2$[28]. In addition, according to the calculation framework proposed by Enapter[45] (which takes full account of the operating and maintenance costs over the lifetime if the electrolyzer is used on a large-scale), the total hydrogen levelized cost of our electrolyzer is estimated to be €3.79 per kg of H$_2$, which is much lower than the projected price for the mass-produced EL Model T in 2023/2024 of €4.15 per kg of H$_2$ (details in Supplementary Note 3). The electrolyzer could further reach a higher current density (500 mA cm$^{-2}$, 65 °C) at the voltage of 1.72 V, indicating the high performance of the asymmetric electrolyzer for water electrolysis (Supplementary Fig. 31). Figure 4g summarizes the comparation of the operating voltage of the state-of-the-art seawater electrolyzers for overall seawater electrolysis at 100 mA cm$^{-2}$ and 25 °C[15,17–27]. In addition, our electrolyzer requires an about 25.3% less in electrical energy consumption at 400 mA cm$^{-2}$ compared to the reported direct seawater electrolysis, which combines seawater purification and subsequent electrolysis[46]. It is obvious that our sodium ions conducted asymmetric electrolyzer with Pt$_{SA}$-Ni$_{6.6}$Fe$_{0.4}$P$_3$ and Ni$_5$Fe$_2$P$_3$ as the catalysts achieves the lowest energy consumption. To further verify the stability of the asymmetric electrolyzer for direct seawater electrolysis, we applied turbid sea salt water containing Ca$^{2+}$ and Mg$^{2+}$ in the cathode chamber with the one-way flowing feed scheme to maintain the electrolyte pH (Supplementary Fig. 32). The electrolyzer demonstrates relatively stable voltage response at 100 mA cm$^{-2}$. Moreover, white precipitates are not observed on both the electrodes and the membrane after 14 h. These phenomena further certify the stability of the electrolyzer and the catalysts therein.

The changes in structure and catalytic performance of Pt$_{SA}$-Ni$_{6.6}$Fe$_{0.4}$P$_3$ and Ni$_5$Fe$_2$P$_3$ after the stability test were also studied (Supplementary Figs. 33–35). The XRD of Pt$_{SA}$-Ni$_{6.6}$Fe$_{0.4}$P$_3$ after stability test shows no typical peaks for Mg(OH)$_2$ and Ca(OH)$_2$, indicating

almost no Ca$^{2+}$ or Mg$^{2+}$ precipitates forms under the effect of flow seawater during continuous operation. Moreover, they both demonstrate negligible morphology change and activity degradation, further indicating their excellent stability. In addition, the elemental composition of the catalysts after 24 h stability test has not significantly changed (Supplementary Table 3). The mass ratio of Pt in Pt$_{SA}$-Ni$_{6.6}$Fe$_{0.4}$P$_3$ is nearly unchanged (1.32 vs. 1.35 wt%), confirming the high stability of the catalysts. The anode electrolyte was also analyzed to prove that the Na$^+$ exchange membrane can prevent the transport of Cl$^-$ to the anode. Ion chromatography was used to detected specific concentration of Cl$^-$ in the anode electrolyte with different membranes (Fig. 4h and Supplementary Fig. 36). The concentration of Cl$^-$ of the anode electrolyte with Na$^+$ exchange membrane is only 0.0011 M, while that with anion exchange membrane is 0.1964 M, which proves the ability of the Na+ exchange membrane to prevent Cl$^-$ crossover. As shown in the inset of Fig. 4h (details in Supplementary Fig. 37), large amount of white AgCl precipitate appeared in the anode electrolyte after 12 h electrolysis for the electrolyzer with commercial anion exchange membrane when adding AgNO$_3$, while brown precipitate that corresponding to Ag$_2$O appears in the anode electrolyte for asymmetric electrolyzer with Na$^+$ exchange membrane, further indicating that almost no Cl$^-$ ions crossed the Na$^+$ exchange membrane to the anode. Therefore, side reactions such as ClOR and the potential corrosion of Cl$^-$ on the anode[5] can be diminished.

## Mechanism study

Our previous work has proved that amorphous Ni-Fe-P is an efficient OER catalyst due to the incorporation of phosphorus atoms that breaks the scaling relations of adsorbed OER intermediates and facilitates the OER kinetics[13]. To further understand the OER performance of amorphous Ni$_5$Fe$_2$P$_3$ NWs in this work, in situ Raman spectroscopy of Ni$_5$Fe$_2$P$_3$ was investigated at OER-relevant potential region (1.2 to 1.5 V$_{RHE}$) (Supplementary Fig. 38 and Supplementary Note 4). The peaks observed at 1.4 V$_{RHE}$ can be attributed to NiOOH, indicating that the surface reconstruction of amorphous Ni$_5$Fe$_2$P$_3$ to metal hydroxide during OER test. The hydroxide with remained P would contribute to the excellent OER performance of Ni$_5$Fe$_2$P$_3$[13]. The in situ Raman spectroscopy of Pt$_{SA}$-Ni$_{6.6}$Fe$_{0.4}$P$_3$ at the potential window of −0.5 to −0.2 V$_{RHE}$ in NaCl solution, where HER is expected to take place, was also carried out to elucidate the mechanism for the catalyst enhanced HER activity. The extensive band from 3000 cm$^{-1}$ to 3800 cm$^{-1}$ in Fig. 5a, b is ascribed to the O-H stretching mode of interfacial water ($v_{O-H}$), which is sensitive to the chemical environment[47]. The $v_{O-H}$ band can be separated into three Gaussian peaks as suggested, corresponding to three types of O-H stretching vibrations[48]. The low wavenumber component (Fig. 5a, b, blue) and middle component (Fig. 5a, b, orange) are ascribed to 4-coordinated hydrogen-bonded water and 2-coordinated hydrogen-bonded water, respectively. The high wavenumber component (Fig. 5a, b, purple) is attributed to the Na$^+$ hydrated water with weak hydrogen-bonded interaction[49]. The peak area of Na$^+$ hydrated water increases as the potential decreases for Pt$_{SA}$-Ni$_{6.6}$Fe$_{0.4}$P$_3$ compared to Pt foil, indicating the interfacial water structure could be reoriented to more beneficial conditions for water dissociation at negative potentials for Pt$_{SA}$-Ni$_{6.6}$Fe$_{0.4}$P$_3$, which may be ascribed to the effect of P as the electron acceptor[50,51]. The first step of HER in unbuffered neutral media is generally considered to be a slow kinetic water molecule dissociation process[52]. Therefore, the promoted water dissociation process on Pt$_{SA}$-Ni$_{6.6}$Fe$_{0.4}$P$_3$ would contribute to its improved HER performance. In addition, the band at 1600 cm$^{-1}$ is ascribed to the H-O-H bending mode ($\delta_{H-O-H}$) of the interfacial water, and the band at 2030 cm$^{-1}$ to 2100 cm$^{-1}$ is attribute to the vibration mode of H atoms coordinated on the top of the surface Pt atom ($v_{Pt-H}$)[53]. As shown in Fig. 5a, b, the band of $\delta_{H-O-H}$ exists at −0.2 V$_{RHE}$ while the band of $v_{Pt-H}$ is absent for both Pt$_{SA}$-Ni$_{6.6}$Fe$_{0.4}$P$_3$ and Pt foil, indicating that the first

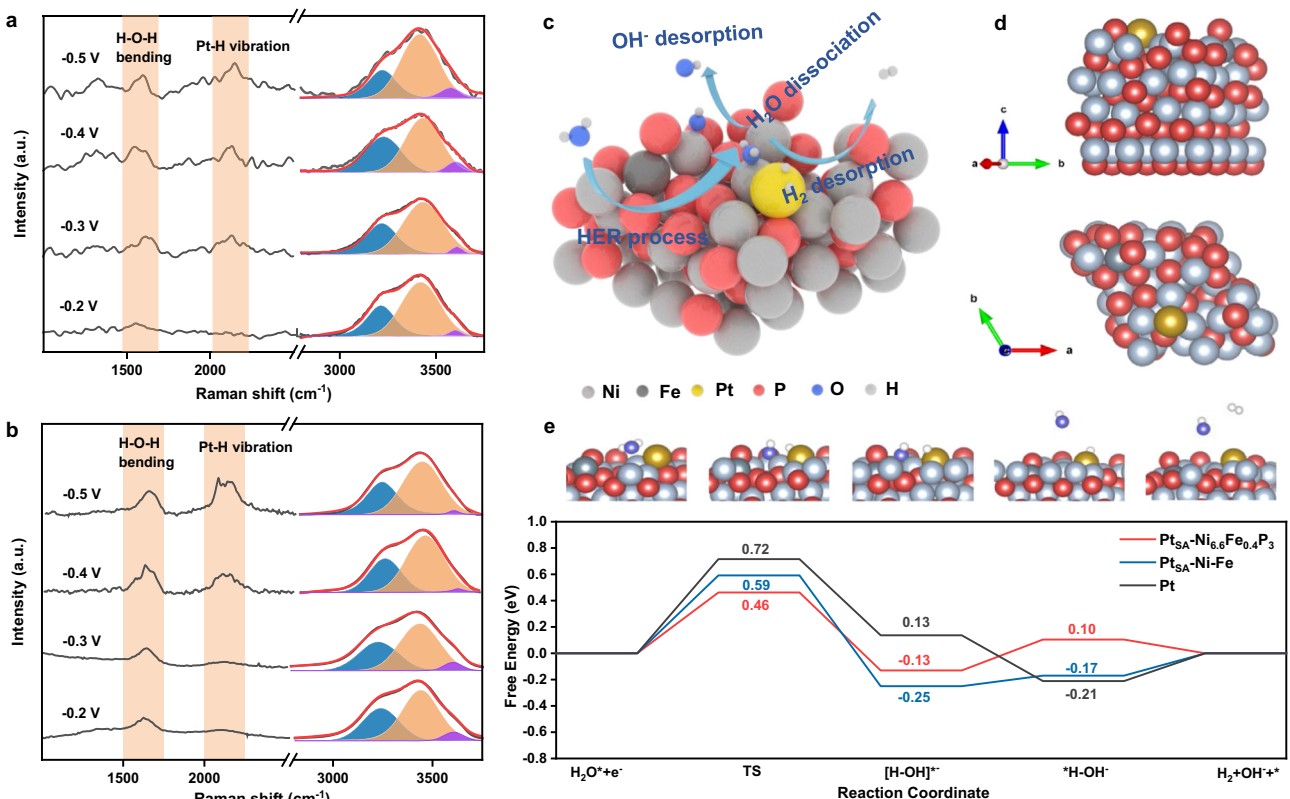

**Fig. 5 | Mechanism study. a** The in situ Raman spectroscopy in NaCl solution at the surface of $Pt_{SA}$-$Ni_{6.6}Fe_{0.4}P_3$. **b** The in situ Raman spectroscopy in NaCl solution at the surface of Pt foil. **c** Mechanism of $Pt_{SA}$-$Ni_{6.6}Fe_{0.4}P_3$ for HER in NaCl solution. **d** Computational model of $Pt_{SA}$-$Ni_{6.6}Fe_{0.4}P_3$. **e** Free energy diagram of HER in NaCl solution on these three catalyst surfaces. TS represents the activated $H_2O$ dissociation energy of the transition state. Top: Structural configurations of various states on the $Pt_{SA}$-$Ni_{6.6}Fe_{0.4}P_3$ surface during the HER reaction. (gray: nickel, dark gray: iron, red: phosphorus, yellow: platinum, blue: oxygen, white: hydrogen).

adsorption layer are water molecules at this potential. However, the band of $\nu_{Pt\text{-}H}$ appears at $-0.3\ V_{RHE}$ for $Pt_{SA}$-$Ni_{6.6}Fe_{0.4}P_3$ catalyst, which is more positive than that of the Pt ($-0.4\ V_{RHE}$), further revealing that amorphous $Pt_{SA}$-$Ni_{6.6}Fe_{0.4}P_3$ may benefit the dissociation of the adsorbed water and exhibit enhanced HER activity.

Based on the analysis of in situ Raman spectroscopy and EXAFS, we propose that water molecules are first adsorbed at the Ni sites adjacent to Pt atoms, followed by their dissociation and the formation of adsorbed H at the Pt sites. Finally, the adsorbed H atoms at Pt sites are released and $H_2$ is formed (Fig. 5c). Density functional theory (DFT) calculations were further investigated by an implicit solvation calculation to disclose the enhanced HER kinetics on the amorphous $Ni_{6.6}Fe_{0.4}P_3$ supported Pt single atoms in NaCl solution which can consider the solvent effect of the solvents and ion solvation[54]. The models of $Pt_{SA}$-$Ni_{6.6}Fe_{0.4}P_3$, $Pt_{SA}$-Ni-Fe and Pt foil were built based on the structural characterization in previous section (Fig. 5d and Supplementary Fig. 39, details in Supplementary Note 5). The energy barrier of HER in NaCl on various catalysts includes two key steps: the water dissociation and H* desorption (Supplementary Fig. 40 and Supplementary Note 6)[32]. As shown in Fig. 5e, the energy barrier of water dissociation, which is the rate-determining step on $Pt_{SA}$-$Ni_{6.6}Fe_{0.4}P_3$ (0.46 eV), is lower than that of Pt foil (0.72 eV). It indicates that $Pt_{SA}$-$Ni_{6.6}Fe_{0.4}P_3$ could facilitate the water dissociation, which is consistent with the in situ Raman spectroscopy. The $Pt_{SA}$-Ni-Fe has a higher energy barrier of water dissociation (0.59 eV) compared to $Pt_{SA}$-$Ni_{6.6}Fe_{0.4}P_3$, indicating that the amorphous $Ni_{6.6}Fe_{0.4}P_3$ formed after P-doping is the key structure to promote the water dissociation process. Moreover, the free energy of desorption of H* on $Pt_{SA}$-$Ni_{6.6}Fe_{0.4}P_3$ (0.10 eV) is closer to zero compared with that on $Pt_{SA}$-Ni-Fe ($-0.17$ eV) and Pt foil ($-0.21$ eV), indicating a faster desorption of H*. The lower

energy barrier of water dissociation and a faster desorption of H* account for the best HER performance of $Pt_{SA}$-$Ni_{6.6}Fe_{0.4}P_3$.

## Discussion

In summary, we demonstrate a pH-asymmetric electrolyzer using a $Na^+$ exchange membrane as the separator and amorphous Ni-Fe-P NWs as the main catalysts to realize direct seawater electrolysis under low voltage. The $Na^+$ exchange membrane can prevent $Cl^-$ from passing through to anode, thus avoiding the undesired ClOR, while the difference of chemical potentials between the cathode and anode electrolytes can be harvested to reduce the hydrogen production energy cost. The issue related to the formation of $Mg^{2+}$ and $Ca^{2+}$ precipitates during seawater electrolysis is also greatly alleviated in this system. The superhydrophilic $Pt_{SA}$-$Ni_{6.6}Fe_{0.4}P_3$ catalysts on porous Ni foam with lower adhesive force to the bubbles results in fast bubble release from the electrode surface, thus accelerating the mass transport. In situ Raman spectroscopy and DFT calculation reveals that water dissociation is promoted on $Pt_{SA}$-$Ni_{6.6}Fe_{0.4}P_3$ (energy barrier decreases from 0.72 eV on Pt to 0.46 eV), thus reducing the electrochemical polarization for HER in neutral media. In this way, the asymmetric electrolyzer exhibits the current densities of 10 mA cm$^{-2}$ and 100 mA cm$^{-2}$ at low voltages of 1.31 and 1.46 V for direct seawater electrolysis, respectively. It also reaches a current density of 400 mA cm$^{-2}$ at low voltage of 1.66 V at 80 °C, corresponding to the electricity price of US\$1.36 per kg of $H_2$, which is lower than the DOE 2025 target of US\$1.4 per kg of $H_2$. The total hydrogen levelized cost is also estimated to be €3.79 per kg of $H_2$ according to the calculation framework proposed by Enapter, which is much lower than the projected price for the mass-produced EL Model T in 2023/2024 of €4.15 per kg of $H_2$. This work provides an efficient electrolyzer architecture

and catalyst design route for direct seawater electrolysis with low energy cost.

## Methods

### Materials

Ammonium chloride ($NH_4Cl$, 99.5%), nickel sulfate hexahydrate ($NiSO_4 \cdot 6H_2O$, 99%), iron sulfate heptahydrate ($FeSO_4 \cdot 7H_2O$, 99%), diethylene glycol ($C_4H_{10}O_3$, 99%), sodium hydroxide (NaOH, ≥99%), ethanol ($CH_3CH_2OH$, ≥99.5%) and sodium chloride (NaCl, ≥99%) were purchased from Innochem. Ni foam (KunShan-Yierda, China, thickness 0.15 cm), phosphorus (Aladdin, >98.9%, −100 mesh), chloroplatinic acid hexahydrate (Aladdin), seawater (from the Yellow sea, China), ruthenium dioxide ($RuO_2$, Innochem), commercial Pt/C (Johnson Matthey, 20 wt%) and Nafion® 211 membrane were used as received. Deionized (DI) water (resistivity: 18.3 MΩ cm) was used for the preparation of all aqueous solutions.

### Catalyst syntheses

**Synthesis of Ni-Fe alloys.** Ni-Fe/Ni foam anode was prepared by cathodic electrodeposition of porous Ni microsphere arrays on a pre-shaped Ni foam ($0.5 \times 1 cm^2$). Typically, the deposition was performed in a standard two electrode configuration at room temperature with an electrolyte of 2.0 M $NH_4Cl$, 0.095 M $NiSO_4$ and 0.005 M $FeSO_4$. Pre-shaped commercial Ni foam and platinum foil were used as the working electrode and counter electrode, respectively. The electrodeposition was carried out at a constant current of −1.5 A $cm^{-2}$ for 5 min to obtain Ni-Fe/NF. Ni-Fe alloys with different NiFe ratios can be obtained by regulating the ratio of $NiSO_4$ and $FeSO_4$ in the electrolyte.

**Synthesis of $Ni_{6.6}Fe_{0.4}P_3$ NWs.** The $Ni_{6.6}Fe_{0.4}P_3$ NWs were synthesized by a simple solvothermal method. Typically, 600 mg phosphorus was dispersed in 10 mL diethylene glycol and stirred for 30 min. Subsequently, the solution was transferred into 50 mL Teflon-lined autoclave and mixed with the pretreated Ni-Fe foam, maintained at 250 °C for 6 h, later naturally cooled to room temperature. Then, the as-obtained $Ni_{6.6}Fe_{0.4}P_3$ NWs underwent an ethanol-water-ethanol cleaning process for 10 min, respectively.

**Synthesis of $Pt_{SA}$-$Ni_{6.6}Fe_{0.4}P_3$ NWs.** $Pt_{SA}$-$Ni_{6.6}Fe_{0.4}P_3$ was fabricated by the electrochemical reduction process in the three-electrode system, in which the fabricated $Ni_{6.6}Fe_{0.4}P_3$ was performed as the working electrode, graphite sheet acted as a counter electrode, saturated calomel electrode acted as a reference electrode. The corresponding electrochemical process was carried out by multi-cycle cathode polarization in 1 M NaOH solution (100 ml) containing 4 μM $H_2PtCl_6$ with a scan rate of 50 mV $s^{-1}$ between 0 and −0.40 V versus RHE for 800 cycles.

**Synthesis of $Pt_{SA}$-Ni-Fe.** $Pt_{SA}$-Ni-Fe was fabricated by the electrochemical reduction process in the three-electrode system, in which the fabricated Ni-Fe alloy was performed as the working electrode, graphite sheet acted as a counter electrode, saturated calomel electrode acted as a reference electrode. The corresponding electrochemical process was carried out by multi-cycle cathode polarization in 1 M NaOH solution (100 ml) containing 4 μM $H_2PtCl_6$ with a scan rate of 50 mV $s^{-1}$ between 0 and −0.40 V versus RHE for 800 cycles.

**Synthesis of 20 wt% PtC/NF.** The performance of commercial Pt/C were tested after it is supported on Ni foam in this work to keep the close surface area. The ink of 20 wt% commercial Pt/C was covered on Ni foam and the total Pt loading was 60 μg on the Ni foam ($0.5 \times 1 cm^2$).

**Preparation of $Na^+$ ions exchange membrane.** The $Na^+$ ions exchange membrane was obtained by the treatment of commercial Nafion® 211 membrane. The commercial Nafion® 211 membrane was first immersed in 5 vol% $H_2O_2$ at 70 °C for 1 h. After cleaning with deionized water, the membrane was then immersed in 1 M NaOH at 70 °C for 12 h to obtain the $Na^+$ ions exchange membrane.

**Assembly of catalysts in asymmetric electrolyzer.** The catalysts used in asymmetric electrolyzer was the same as that in half-cell with a size of $0.5 \times 1 cm^2$. the Pt loading was about 40 μg (determined by ICP-MS) in the $Pt_{SA}$-$Ni_{6.6}Fe_{0.4}P_3$. Porous catalysts were extruded on the $Na^+$ ions exchange membrane carefully and applied in asymmetric electrolyzer.

### Characterizations

The morphology measurement of the synthesized catalysts was performed by SEM (Nova NanoSEM 450 (FEI, USA)). TEM characterizations were obtained on a Tecnai G2 20 (FEI, USA). HRTEM images, HAADF-STEM images, and STEM-EDS mapping images were obtained by a FEI Tecnai F20. The Pt contents in the catalysts were measured by inductively coupled plasma optical emission spectrometry. XRD patterns were collected from Rigaku MiniFlex 600 diffractometer with a Cu radiation source ($\lambda = 0.15406$ nm. XPS spectra were collected from Thermo Scientific K-Alpha. X-Ray Fluorescence (XRF) results were obtained from M4 TORNADO. Raman measurements were performed using a LabRAM HR Evolution (Horiba JobinYvon, France) with 532 nm excitation wavelength. Extended X-ray absorption fine structure spectroscopy (EXAFS) at the Pt L3-edge was performed at BL11B of Shanghai Synchrotron Radiation Facility.

### Electrochemical measurements

The OER and HER polarization curves were carried out with a CHI 760D (Chenhua, China) electrochemical workstation with a typical three-electrode system at 25 °C, in which fabricated catalysts were directly employed as the working electrode, graphite sheet acted as a counter electrode, saturated calomel electrode (SCE) acted as a reference electrode. The potential was calibrated to reversible hydrogen electrode (RHE) through measuring the potential difference between the SCE and RHE. LSV with 90% iR-compensation were tested under the scan rate of 5 mV $s^{-1}$. The value of iR-compensation was automatically 90% compensated by the CHI 760D. The performance of asymmetric and symmetric electrolyzers were tested on a LAND C3001B battery measurement system (Wuhan, China), and the Chronopotentiometry curves at 10 mA $cm^{-2}$ and 100 mA $cm^{-2}$ were presented with manual 90% iR-compensation. The value of resistance was tested by CHI 760D which was $0.5 \pm 0.05$ Ω. The area of electrodes in the electrolyzer are 0.5 $cm^2$. The flow rate of peristaltic pump is 6 ml $min^{-1}$. The test temperature was controlled by heating the input electrolyte to the specified temperature. The electrochemical active surface area (ECSA) can be estimated with the use of the double-layer capacitances ($C_{dl}$). The specific capacitance for a flat surface ($C_s$) is supposed to be ~ 40 μF $cm^{-2}$, and the ECSA is estimated by the following formula[55]:

$$ECSA = \frac{C_{dl}}{C_s}$$

### XAFS measurements and data processing

Pt L3-edge XAFS measurements were performed at BL11B station in Shanghai Synchrotron Radiation Facility (SSRF). The electron storage ring of SSRF was operated at 3.5 GeV, with a maximum current of 250 mA. XAFS data were collected using a fixed-exit Si(111) double-crystal monochromator, and the energy was calibrated using metals foil. The samples were pelletized as disks of 13 mm diameter with 1 mm thickness by using LiF power as binder. Utilizing the ATHENA module of the IFEFFIT software packages, the obtained EXAFS data were

performed according to the standard procedures[56] (Nucl. Sci. Tech., 2015, 26, 50102). The EXAFS contributions were separated from different coordination shells by using a hanning windows (dk = 1.0 Å$^{-1}$). Fits were carried out using a k range of 3–13.5 Å$^{-1}$ and a R range of 1.1–2.9 Å using the module ARTEMIS of IFEFFIT. the overall amplitude reduction factor $S_0^2$ was fixed to the best-fit value of 0.81 determined from fitting the data of metal Pt foil.

## In situ Raman experimental setup

The electrochemical Raman measurements were carried out on Lab-RAM HR800. A Nd-YAG laser with 532 nm excitation wavelength and a 50× microscope objective with a numerical aperture of 0.5 were used in all measurements. Raman frequency was calibrated by a Si wafer during each experiment. In situ electrochemical Raman experiments were employed in a C031 in situ Raman cell from Gaoss Union and a CHI 760D (Chenhua, China) electrochemical workstation was used to control the potential. Each spectrum was obtained with the exposure time of 60 s and accumulating twice.

## DFT calculation details

Theoretical calculations: The Vienna Ab Initio Package (VASP) was employed to perform all the density functional theory (DFT) calculations within the generalized gradient approximation (GGA) using the Perdew, Burke, and Enzerhof (PBE) formulation[57–59]. The projected augmented wave (PAW) potentials were applied to describe the ionic cores and take valence electrons into account using a plane wave basis set with a kinetic energy cutoff of 450 eV[60]. Partial occupancies of the Kohn–Sham orbitals were allowed using the Gaussian smearing method and a width of 0.05 eV. The electronic energy was considered self-consistent when the energy change was smaller than 10$^{-6}$ eV. A geometry optimization was considered convergent when the force change was smaller than 0.03 eV/Å. Grimme's DFT-D3 methodology was used to describe the dispersion interactions[61]. For implicit solvation calculations, we used VASPsol[54], a software package that incorporates solvation into VASP within a self-consistent continuum model. VASPsol, due to its simplicity and low computational costs. The energy from DFT is added to the energies from electrostatic interactions between the solute and the solvent and the cavitation energy to create the solute within the solvent. The vacuum spacing perpendicular to the plane of the structure is 20 Å. The Brillouin zone integral utilized the surfaces structures of 2 × 2 × 1 Monkhorst–Pack K-point sampling. The free energy was calculated using the equation:

$$G = E_{ads} + ZPE - TS$$

where $G$, $E_{ads}$, ZPE, and TS are the free energy, total energy from DFT calculations, zero-point energy and entropic contributions, respectively.

## Data availability

The experimental data generated in this study have been provided in the Supplementary information. Any additional data are available from the corresponding author.

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

## Acknowledgements

This work was financially supported by the National Key Research and Development Program of China (2021YFA1600800 to Q.L. and T.W.), National Natural Science Foundation of China (22122202 to Q.L., 22072051 to T.W., 21972051 to Q.L.), the Natural Science Foundation of Hubei Province (2021CFB329 to T.W.). The authors thank the Analytical and Testing Center of Huazhong University of Science and Technology (HUST) for carrying out the SEM, XPS, and XRD measurements.

## Author contributions

Q.L. and T.W. supervised the project and conceived the idea. H.S. performed sample synthesis and electrochemical characterizations. X.L. and Z.C. conducted theoretical calculations. J. Liu and S. Liu performed Raman characterizations. S. Li and J. Liang helped in XAS characterization. W.C., D.S., and C.W. contributed to TEM and HAADF-STEM characterizations. H.S, T.W., and Q.L. wrote the manuscript. Y.H. and L.E. participated in discussing the data and revising the manuscript.

## Competing interests

The authors declare no competing interests.
