## [Peer Review File · Nature Communications]

Reviewer comments, first round review -

Reviewer #1 (Remarks to the Author):

In this work, the authors reported an interesting pH-asymmetric electrolyzer with Na⁺ ions exchange membrane for direct seawater electrolysis. The system with PtSA-Ni_{6.6}Fe_{0.4}P₃ and Ni₅Fe₂P₃ could reach the current density of 10 mA cm⁻² and 100 mA cm⁻² at exceptionally low voltages of 1.31 V and 1.46 V. Besides, the key problems in direct seawater electrolysis (Cl⁻ corrosion, Mg²⁺ and Ca²⁺ precipitates) can be well addressed in this innovative asymmetric electrolyzer and the origin of the catalytic performance was demonstrated by the DFT calculation and in-situ Raman spectroscopy. This work realized high efficiency and low-energy consumption strategy for direct seawater splitting to produce hydrogen and solved the key problems in the direct seawater electrolysis field. Overall this paper is systematically well organized with new insights for developing efficient and durable electrocatalysts in seawater electrolyzer, and I believe that the manuscript could be accepted after minor revision. Comments for this article are attached as follows:

(1) The asymmetric electrolyzer with Na⁺ ions exchange membrane exhibited lower voltage than the symmetric electrolyzer at the same current density. However, a previous report suggests that the symmetric feed performed better than asymmetric feed with anion exchange membrane (Energy Environ. Sci., 2020, 13, 1725). So the authors should provide more discussions to reveal the beneficial properties of the asymmetric electrolyzer in this work.

(2) OH⁻ ions will be produced by the water dissociation on the cathode electrode surface in neutral media, thus the local pH of the cathode electrode will gradually increase and the Ca²⁺ and Mg²⁺ precipitates may form on the surface of electrodes. But in this work, the problem of Ca²⁺ and Mg²⁺ precipitates seem to be partly solved. The author should make more explanations.

(3) The asymmetric electrolyzer performed the current density of 100 mA cm⁻² at the voltage of 1.50 V for real seawater while 1.46 V for NaCl solution. The author should explain why the performance was slightly reduced than that for NaCl solution.

(4) The authors should provide the elemental composition of the catalysts after stability test to demonstrate the stability of the catalysts.

(5) The peaks of the in-situ Raman spectroscopy in Supplementary Fig. 32 should be attributed to NiOOH (Angew. Chem. Int. Ed. 2021, 60, 19774; Angew. Chem. Int. Ed. 2019, 58, 1252.). The authors should confirm it carefully.

Reviewer #2 (Remarks to the Author):

The manuscript by Li et al. reported sodium-ion-conducted asymmetric water splitting electrolyzer for H₂ production and demonstrated a low cell voltage of 1.46 V at 100 mA/cm². They further achieved a low cell voltage of 1.46 V at 400 mA/cm² at 800 oC, corresponding to the energy price of \$1.36 per kg of H₂. In their cell, Pt on amorphous Ni_{6.6}Fe_{0.4}P₃ nanowires was used as the anode while Ni₅Fe₂P₃ was used as the cathode. Density functional theory calculations and in situ Raman spectroscopy were used to explore the fundamental mechanism of Pt on Ni_{6.6}Fe_{0.4}P₃ nanowires for hydrogen evolution. The energy price of \$1.36 per kg of H₂ is impressive. However, the calculation of energy price is not convincing. Some other points also need to be addressed before this work can be published. Detailed points are outlined below.

1. The energy price of H₂ in Supplementary note 2 is not fair at all. The authors only included the energy consumption and electricity bill. Per department of energy's guideline, the calculation at least needs to consider the Capex, Opex, lifetime of the electrolyzer/catalysts, efficiency of the electrolyzer. <https://www.enapter.com/newsroom/h2-view-exclusive-calculating-the-cost-of->

green-hydrogen.

2. A few publications have shown the direct seawater electrolysis for hydrogen production, even at current density higher than 500 mA/cm². The authors should clearly clarify the novelty for publishing the paper on high-impact journals like Nature Communications. Examples of previous papers are here. Nature volume 612, pages673-678 (2022); Nature Communications volume 12, Article number: 4182 (2021); ACS Energy Lett. 2019, 4, 4, 933-942;

3. The authors designed a pH-asymmetric electrolyzer for direct seawater electrolysis. Can the pH maintain on both anode and cathode side during the long operation? The authors should check the pH after long-time cycling. Also, would the local pH near the electrode be different under operating conditions?

4. The evidence of Pt single atoms is lacking as well. The authors performed Pt L3-edge EXFAS to prove the coordination of Pt-Ni. How would the Pt L3-edge be fitted if assuming the coordination is Pt-Fe? EXAFS fitting can be tricky and the difference between Pt-Ni and Pt-Fe is not obvious. Also, for single atom evidence, HAADF-STEM to see the distribution of Pt over Ni_{6.6}Fe_{0.4}P₃ is a more direct evidence.

5. The stability of the asymmetric electrolyzer at constant current density of 400 mA cm⁻², 80 °C without iR-compensation in Supplementary Figure 24 decays quick fast. Some explanation is needed, especially using the initial cell voltage for H₂ price calculation is not fair or acceptable at all.

6. Did the authors consider the electrolyte when calculating the free energy diagram of HER on different catalysts? Solvents and ion solvation plays an important role in influencing the energetics.

Itemized list of response to the editor and reviewers' remarks

Reviewer #1:

In this work, the authors reported an interesting pH-asymmetric electrolyzer with Na⁺ ions exchange membrane for direct seawater electrolysis. The system with Pt_{SA}-Ni_{6.6}Fe_{0.4}P₃ and Ni₅Fe₂P₃ could reach the current density of 10 mA cm⁻² and 100 mA cm⁻² at exceptionally low voltages of 1.31 V and 1.46 V. Besides, the key problems in direct seawater electrolysis (Cl⁻ corrosion, Mg²⁺ and Ca²⁺ precipitates) can be well addressed in this innovative asymmetric electrolyzer and the origin of the catalytic performance was demonstrated by the DFT calculation and in-situ Raman spectroscopy. This work realized high efficiency and low-energy consumption strategy for direct seawater splitting to produce hydrogen and solved the key problems in the direct seawater electrolysis field. Overall this paper is systematically well organized with new insights for developing efficient and durable electrocatalysts in seawater electrolyzer, and I believe that the manuscript could be accepted after minor revision.

Reply: Thank the reviewer for the positive comments on our work.

Comment 1: The asymmetric electrolyzer with Na⁺ ions exchange membrane exhibited lower voltage than the symmetric electrolyzer at the same current density. However, a previous report suggests that the symmetric feed performed better than asymmetric feed with anion exchange membrane (Energy Environ. Sci., 2020, 13, 1725). So the authors should provide more discussions to reveal the beneficial properties of the asymmetric electrolyzer in this work.

Reply: Thank you for these suggestions. In this asymmetric electrolyzer with Na⁺ exchange membrane using a flow-cell model, the electrolyte flows through the channels of the porous electrodes, thus the pH at the cathode is near Neutral, and the pH at the anode, where the Ni₅Fe₂P₃ is, is about pH = 14. In this case, the difference in chemical potential could be leveraged to reduce the voltage for water splitting to 0.816 V, 0.414 V lower than the voltage measures with the mentioned symmetric system in alkaline conditions (Figure 1b). Even though Pt_{SA}-Ni_{6.6}Fe_{0.4}P₃ demonstrates inferior HER performance in neutral solution without a buffer ($\eta \approx 408$ mV at 100 mA/cm²) when compared with its performance in 1 M NaOH ($\eta \approx 89$ mV at 100 mA/cm²), a reduced operating voltage of about 0.1 V could be achieved with the later system. Therefore, the Na⁺ based asymmetric electrolyzer developed in this work displays better performance compared with the mentioned symmetrical electrolyzer. In contrast, the asymmetric electrolyzer with anion exchange membrane (Energy Environ. Sci., 2020, 13, 1725) may not work that way, because hydroxide ions can easily diffuse to the cathode to affect the pH of the cathode-catalyzed reaction. Thus,

the chemical energy change that is promoted by the pH difference between the cathode and anode electrolytes may not fully leveraged to increase the thermodynamic efficiency of the process. Furthermore, Pt/C catalysts were used as the cathode catalysts in this work (Energy Environ. Sci., 2020, 13, 1725). However, the difference in the HER performance of Pt/C catalysts in alkaline and neutral unbuffered solutions ($\eta \approx 573$ mV at 100 mA cm^{-2} , tested in our work) is greater than the compensation value for the pH difference (0.414 mV), which is another reason for the higher voltage measured with a asymmetric system that work (Energy Environ. Sci., 2020, 13, 1725).

Comment 2: OH^- ions will be produced by the water dissociation on the cathode electrode surface in neutral media, thus the local pH of the cathode electrode will gradually increase and the Ca^{2+} and Mg^{2+} precipitates may form on the surface of electrodes. But in this work, the problem of Ca^{2+} and Mg^{2+} precipitates seem to be partly solved. The author should make more explanations.

Reply: Thanks for noticing this important attribute of our system and for the valuable and insightful comments. Although the OH^- would be produced by the water dissociation on the cathode surface during the HER process, the calcium and magnesium deposits growth are strongly correlated with the hydrodynamic conditions (Environ Chem Lett, 2020, 18, 1193). In high flow rates, the OH^- ions produced are quickly evacuated from the interface and the interfacial pH is not high enough for precipitation (pH 9.5 and 13 are required for $\text{Mg}(\text{OH})_2$ and $\text{Ca}(\text{OH})_2$, respectively) (Electrochim Acta 2000, 45:1837–1845; RSC Adv., 2021,11, 8306-8313). Thus, the formation of calcium and magnesium precipitates will be inhibited to some extent at high flow rates used in our system. The HER stability test of $\text{Pt}_{\text{SA}}\text{-Ni}_{6.6}\text{Fe}_{0.4}\text{P}_3$ in natural seawater at 100 mA cm^{-2} and 600 rpm magnetic stirring was studied (**Figure R1a** and new **Figure S15**). The stable response indicates the Ca^{2+} and Mg^{2+} will not have a significant effect on the catalyst and electrolyzer performance. The XRD of $\text{Pt}_{\text{SA}}\text{-Ni}_{6.6}\text{Fe}_{0.4}\text{P}_3$ after stability test shows no typical peaks for $\text{Mg}(\text{OH})_2$ and $\text{Ca}(\text{OH})_2$, indicating almost no Ca^{2+} and Mg^{2+} precipitates formed during continuous operation.(**Figure R1b** and new **Figure S33**).

Figure R1 (a) The chronopotentiometry curves of $\text{Pt}_{\text{SA}}\text{-Ni}_{6.6}\text{Fe}_{0.4}\text{P}_3$ at 100 mA cm^{-2} and the photograph of the $\text{Pt}_{\text{SA}}\text{-Ni}_{6.6}\text{Fe}_{0.4}\text{P}_3$ after stability test in seawater (inset). (b) The XRD pattern of $\text{Pt}_{\text{SA}}\text{-Ni}_{6.6}\text{Fe}_{0.4}\text{P}_3$ after stability test.

The following discussion has been added in the revised manuscript in the part relating to the electrocatalytic performance of seawater electrolysis:

“The XRD of $\text{Pt}_{\text{SA}}\text{-Ni}_{6.6}\text{Fe}_{0.4}\text{P}_3$ after stability test shows no typical peaks for $\text{Mg}(\text{OH})_2$ and $\text{Ca}(\text{OH})_2$, indicating almost no Ca^{2+} or Mg^{2+} precipitates forms under the effect of flow seawater during continuous operation.”

Comment 3: The asymmetric electrolyzer performed the current density of 100 mA cm^{-2} at the voltage of 1.50 V for real seawater while 1.46 V for NaCl solution. The author should explain why the performance was slightly reduced than that for NaCl solution.

Reply: Thanks for the valuable and insightful comment. The reason for the difference between the voltage in seawater and NaCl solution may be the small pH difference between natural seawater and NaCl solution. The pH of natural seawater was measured as 8.1, while that of 4 M NaCl solution was 6.9. The pH difference between these may cause small differences in voltage compensation values, which in turn causes a slight increase in the voltage for natural seawater.

Comment 4: The authors should provide the elemental composition of the catalysts after stability test to demonstrate the stability of the catalysts.

Reply: Thanks for the useful advice. The elemental composition of the catalysts after 24 h stability test is shown in **Table R1** and new **Table S3**. The proportion of its elements do not change significantly compared to the initial ratios, indicating the stability of $\text{Pt}_{\text{SA}}\text{-Ni}_{6.6}\text{Fe}_{0.4}\text{P}_3$ and $\text{Ni}_5\text{Fe}_2\text{P}_3$. The mass fraction of Pt in $\text{Pt}_{\text{SA}}\text{-Ni}_{6.6}\text{Fe}_{0.4}\text{P}_3$ (1.32 wt%) after stability test were also confirmed by ICP-MS, indicating the firm

anchor of Pt atoms.

Table R1. The percentages of atoms of catalysts after stability test.

Catalyst	Pt at%	Ni at%	Fe at%	P at%
Pt _{SA} -Ni _{6.6} Fe _{0.4} P ₃	0.4	66.4	4.0	29.2
Ni ₅ Fe ₂ P ₃	/	51.2	20.3	28.5

The following discussion has been added to the “Electrocatalytic performance for seawater electrolysis” section in the revised manuscript:

“In addition, the elemental composition of the catalysts after 24 h stability test has not significantly changed (Supplementary Table 3). The mass ratio of Pt in Pt_{SA}-Ni_{6.6}Fe_{0.4}P₃ is nearly unchanged (1.32 vs. 1.35 wt%), confirming the high stability of the catalysts.”

Comment 5: The peaks of the in-situ Raman spectroscopy in Supplementary Fig. 32 should be attributed to NiOOH (Angew. Chem. Int. Ed. 2021, 60, 19774; Angew. Chem. Int. Ed. 2019, 58, 1252.). The authors should confirm it carefully.

Reply: Thank you for the valuable comment. After careful confirmation of the reference of the Raman spectroscopy of NiFe (J. Am. Chem. Soc. 2013, 135, 12329–12337; Angew. Chem. Int. Ed. 2019, 58, 1252.), we confirm that the Raman peaks at ≈ 473 and 551 cm^{-1} can be attributed to the e_g bending vibration and the A_{1g} stretching vibration of Ni–O in NiOOH, and the relative intensities (I_{473}/I_{551}) will be influenced by the Fe content. Therefore, we have carefully modified the analysis and corrected the discussion of Raman spectroscopy of Ni₅Fe₂P₃ at OER section in this work (**Figure R2 and new Figure S38**).

Figure R2. The in-situ Raman spectroscopy of Ni₅Fe₂P₃ at OER region (1.2 V_{RHE} to 1.5 V_{RHE}) in 1

M NaOH.

The following discussion has been corrected in the revised manuscript:

“in-situ Raman spectroscopy of Ni₅Fe₂P₃ was investigated at OER-relevant potential region (1.2 to 1.5 V_{RHE}) (Supplementary Fig. 38 and Note 4). The peaks observed at 1.4 V_{RHE} can be attributed to NiOOH, indicating that the surface reconstruction of amorphous Ni₅Fe₂P₃ to metal hydroxide during OER test.”

Reviewer #2:

The manuscript by Li et al. reported sodium-ion-conducted asymmetric water splitting electrolyzer for H₂ production and demonstrated a low cell voltage of 1.46 V at 100 mA/cm². They further achieved a low cell voltage of 1.66 V at 400 mA/cm² at 800 °C, corresponding to the energy price of \$1.36 per kg of H₂. In their cell, Pt on amorphous Ni_{6.6}Fe_{0.4}P₃ nanowires was used as the anode while Ni₅Fe₂P₃ was used as the cathode. Density functional theory calculations and in situ Raman spectroscopy were used to explore the fundamental mechanism of Pt on Ni_{6.6}Fe_{0.4}P₃ nanowires for hydrogen evolution. The energy price of \$1.36 per kg of H₂ is impressive. However, the calculation of energy price is not convincing. Some other points also need to be addressed before this work can be published. Detailed points are outlined below.

Reply: We would like to thank the reviewer for the positive and constructive comments on our work.

Comment 1: The energy price of H₂ in Supplementary note 2 is not fair at all. The authors only included the energy consumption and electricity bill. Per department of energy's guideline, the calculation at least needs to consider the Capex, Opex, lifetime of the electrolyzer/catalysts, efficiency of the electrolyzer.

<https://www.enapter.com/newsroom/h2-view-exclusive-calculating-the-cost-of-green-hydrogen>.

Reply: Thanks for the valuable and insightful comments. We admit that the Capex, Opex and lifetime have not fully been considered during the calculation. However, it should be emphasized that the cost of H₂ produced by water splitting may vary due to the different calculation methods and electricity prices. In the revised manuscript, we re-calculated the energy price of H₂ based on the scenario proposed by Enapter (recommended reference by the reviewer). Here are two methods of our calculations for the energy cost:

(1) It is difficult to accurately assess its Capex of the lab-scale seawater electrolysis system. Based on the fact that our electrolyzer and catalysts are conventional materials, we assume the Capex of EL Model T in 2023/2024 as the

Capex for the asymmetric electrolyzer (€1.4 /kg H₂). And the Opex (€2.39 /kg H₂) of the electrolyzer consists of the electricity cost (€2.2 /kg H₂, electricity bill of €0.05 / kWh), water cost (€0.01 /kg H₂, the water purification cost could be saved in this system) and maintenance cost (€0.18 /kg H₂). And the whole cost of green hydrogen production of the asymmetric electrolyzer for seawater electrolysis is pre-estimated to be €3.79 /kg H₂, which is lower than the EL Model T (€4.15 /kg H₂) in 2023/2024.

(2) We estimate the cost of hydrogen production of the asymmetric electrolyzer based on the proportion of the overall cost of electricity (60.3% for the electricity of €0.05 / kWh, according to Enapter). The calculated result is €3.65 /kg H₂, which is also much lower than the EL Model T (€4.15 /kg H₂) in 2023/2024.

In addition, we have also more carefully evaluated the energy price of H₂ based on DOE guidelines

(https://www.energy.gov/sites/prod/files/2015/06/f23/fcto_myRDD_production.pdf), even though there are reported lab-scale electrolyzers that simply calculate the electricity cost as the cost of hydrogen production (such as Nat. Nanotechnol. 2021, 16, 1371). According to the DOE target, the total hydrogen levelized cost is \$2 /kg H₂, where the feedstock cost contribution (electricity cost and water cost) is \$1.4 /kg H₂ (the price of electricity is \$0.031 /kWh). Specifically, based on the same electricity cost, the feedstock cost in our work is \$1.36 /kg H₂ due to high H₂ production efficiency, which is still lower than that of DOE target. We further supplement the total hydrogen levelized cost of our asymmetric electrolyzer by the way of adding the other cost (capital cost contribution, fixed O&M cost contribution and other valuable cost contribution) according to the DOE target and the calculated value is \$1.96 /kg H₂, which is also lower than the DOE target of \$2 /kg H₂.

Generally speaking, for a hybrid site with both solar and wind, the electricity consumption, partially determined by the efficiency, will be the major contributor to the cost of the hydrogen production (ACS Energy Lett. 2023, 8, 3, 1502). Therefore, it is necessary to design the electrolyzer with high efficiency for seawater electrolysis. In our work, the lower heating value (LHV) of H₂ is applied to calculate the efficiencies of Pt_{SA}-Ni_{6.6}Fe_{0.4}P₃ in the asymmetric electrolyzers (the rate of O₂ generation at 400 mA cm⁻² is 6.24 × 10¹⁷ O₂ cm⁻² s⁻¹; the rate of H₂ generation at 400 mA cm⁻² is 1.25 × 10¹⁸ H₂ cm⁻² s⁻¹; LHV of H₂ is 120 kJ g⁻¹ H₂; H₂ power out is 0.501 W cm⁻²). Therefore, the efficiency of the asymmetric electrolyzer operated at 400 mA cm⁻² is calculated to be 75.5% (0.501 W cm⁻²/0.664 W cm⁻²), which is higher than the efficiency of commercial alkaline water electrolysis which range between 63 - 71% (current density: 0.2-0.6 A cm⁻²), indicating that the asymmetric electrolyzer could decrease energy consumption for seawater electrolysis.

The following discussion about of the levelized hydrogen cost has been modified in the “Electrocatalytic performance for seawater electrolysis” and “Discussion” sections in the revised manuscript:

“The electricity cost of the asymmetric electrolyzer is reduced to 3.96 kWh per m³ H₂, corresponding to US\$1.36 per kg of H₂, which is lower than the U.S. DOE 2025 target of US\$1.4 per kg of H₂ and may be further reduced through engineering system architecture. The total hydrogen levelized cost is estimated at US\$1.96 per kg of H₂, slightly lower than U.S. DOE 2025 target of US\$2 per kg of H₂. In addition, according to the calculation framework proposed by Enapter (which takes full account of the operating and maintenance costs over the lifetime if the electrolyzer is used on a large-scale), the total hydrogen levelized cost of our electrolyzer is estimated to be €3.79 per kg of H₂, which is much lower than the projected price for the mass-produced EL Model T in 2023/2024 of €4.15 per kg of H₂ (details in Supplementary Note 3).”

“The total hydrogen levelized cost is also estimated to be €3.79 per kg of H₂ according to the calculation framework proposed by Enapter, which is much lower than the projected price for the mass-produced EL Model T in 2023/2024 of €4.15 per kg of H₂.”

Comment 2: A few publications have shown the direct seawater electrolysis for hydrogen production, even at current density higher than 500 mA/cm². The authors should clearly clarify the novelty for publishing the paper on high-impact journals like Nature Communications. Examples of previous papers are here.

Nature volume 612, pages673-678 (2022);

Nature Communications volume 12, Article number: 4182 (2021);

ACS Energy Lett. 2019, 4, 4, 933-942;

Reply: We thank the reviewer for the valuable comment. In this work, we innovatively proposed the asymmetric electrolyzer with Na⁺ exchange membrane for direct seawater electrolysis which could address the key problems in seawater electrolysis. Firstly, since both HER and OER are pH-sensitive reactions, the chemical potential difference between the anode and cathode electrolyte could be leveraged to increase the system efficiency (Chem. Soc. Rev., 2021, 50, 1495), and the theoretical operation voltage could be reduced from 1.23 to 0.816 V (as shown in Figure 1b). Secondly, in the asymmetric electrolyzer, natural seawater flows in the cathode, while NaOH solution flows in the anode. The Na⁺ exchange membrane is a cation exchange membrane and could prevent Cl⁻ transport through the membrane to the anode, thus fundamentally avoiding the Cl⁻ corrosion on anode catalysts. Thirdly, the cathode electrolyte is a flowing pH-neutral solution while the Mg²⁺ and Ca²⁺ hydroxide precipitates are usually formed at pH higher than 9.5 (Environ. Chem. Lett. 2020, 18,

1193), therefore the known issue of Ca^{2+} and Mg^{2+} precipitation on the cathode could be alleviated. The excellent durability and XRD after stability test of $\text{Pt}_{\text{SA}}\text{-Ni}_{6.6}\text{Fe}_{0.4}\text{P}_3$ in natural seawater indicates that the Ca^{2+} and Mg^{2+} do not precipitate and insignificant effect on the catalyst performance. (Figure R3 and new Figure S33).

Figure R3 (a) The chronopotentiometry curves of $\text{Pt}_{\text{SA}}\text{-Ni}_{6.6}\text{Fe}_{0.4}\text{P}_3$ at 100 mA cm^{-2} and the photograph of the $\text{Pt}_{\text{SA}}\text{-Ni}_{6.6}\text{Fe}_{0.4}\text{P}_3$ after stability test in seawater (inset). (b) The XRD pattern of $\text{Pt}_{\text{SA}}\text{-Ni}_{6.6}\text{Fe}_{0.4}\text{P}_3$ after stability test.

In the previous work (Nature, 2022, 612, 673-678), an in-situ water purification process based on a self-driven phase transition mechanism was used into seawater electrolysis, and the work could achieve direct seawater electrolysis with approximately 1.95 V and 2.3 V at 250 mA cm^{-2} and 400 mA cm^{-2} , corresponding to the energy consumption of $4.6 \text{ kWh Nm}^{-3} \text{ H}_2$ and $5.3 \text{ kWh Nm}^{-3} \text{ H}_2$, respectively. The asymmetric electrolyzer with Na^+ exchange membrane in our work could also achieve the direct seawater, avoiding the Cl^- corrosion and partly solving the problem of Ca^{2+} and Mg^{2+} precipitations. Importantly, through the utilization of pH difference and the rational design of catalyst, the electrolyzer described in our work could achieve 400 mA cm^{-2} at only 1.66 V , corresponding to a lower energy consumption of $3.96 \text{ kWh Nm}^{-3} \text{ H}_2$. Compared with Nature, 2022, 612, 673-678, the energy consumption of the electrolyzer in our work is reduced by 25.3%.

Another previous work (Nature Communications, 2021, 12, 4182) proposed the chlorine-free hybrid seawater splitting coupling hydrazine degradation, which yielded hydrogen with a low electricity expense of $2.75 \text{ kWh per m}^3 \text{ H}_2$ at 500 mA cm^{-2} . However, hydrazine is an energy source which has the high energy density (1.95 MJ Kg^{-1}). We calculated that the amount of hydrazine required to produce $1 \text{ m}^3 \text{ H}_2$ is 22.5 mol . Assuming that this hydrazine is used for complete combustion, it can produce the 14.04 MJ of heat. Even assuming that this heat is used to produce electricity with 50%

efficiency, it can still generate about 1.96 kWh of electricity. Therefore, the energy consumption for the production of 1 m³ H₂ is at least 4.71 kWh if combining the energy of hydrazine with the power consumed by the electrolysis system, which already exceeds the energy required to electrolyze seawater in our work. In addition, we supplied the chronopotentiometry curve of the asymmetric electrolyzer at a higher current density (500 mA cm⁻², 65 °C). the electrolyzer could achieve the current density of 500 mA cm⁻² at a voltage of 1.72 V, corresponding the energy cost of 4.08 kWh Nm⁻³ H₂ (**Figure R4** and new **Figure S31**).

Figure R4. The performance of the asymmetric electrolyzer at constant current density of 500 mA cm⁻², 65 °C without iR-compensation.

The review (ACS Energy Lett. 2019, 4, 4, 933–942) discusses the opportunity and challenge in direct seawater electrolysis. Direct seawater electrolysis can take full advantage of the seaside wind power, solving the problem of unbalanced distribution of water resources. But the authors also suggest Cl⁻ corrosion to be the great challenge for direct seawater electrolysis to produce hydrogen. In our work, a creative asymmetric electrolyzer was designed and applied in direct seawater electrolysis, which fundamentally solves the Cl⁻ corrosion mentioned in the review. Furthermore, the energy savings from the pH difference is manifested in higher system efficiency. Therefore, this work provided an efficient electrolyzer architecture and catalyst design route for direct seawater electrolysis with low energy cost.

The mentioned papers have been properly cited in the revised manuscript as new **Ref. 16** and **46**. The following discussion has been added on the “Electrocatalytic performance for seawater electrolysis” section in the revised manuscript:

“The rotating ring-disk electrode technique was used to quantitatively detect the local pH on the Pt_{SA}-Ni_{6.6}Fe_{0.4}P₃ for HER in NaCl solution (Supplementary Note 2 and

Supplementary Figs. 23-24)⁴⁴. We find that the pH value on the Pt_{SA}-Ni_{6.6}Fe_{0.4}P₃ increases when HER occurs but it is still far lower than the anode pH (14.4), indicating that the chemical potential due to the pH difference can be harvested.”

“The electrolyzer could further reach a higher current density (500 mA cm⁻², 65 °C) at the voltage of 1.72 V, indicating the high performance of the asymmetric electrolyzer for water electrolysis (Supplementary Fig. 31).”

“In addition, our electrolyzer requires an about 25.3% less electrical energy consumption at 400 mA cm⁻² compared to the reported direct seawater electrolysis, which combines seawater purification and subsequent electrolysis⁴⁶.”

“The XRD of Pt_{SA}-Ni_{6.6}Fe_{0.4}P₃ after stability test shows no typical peaks for Mg(OH)₂ and Ca(OH)₂, indicating almost no Ca²⁺ or Mg²⁺ precipitates forms under the effect of flow seawater during continuous operation.”

Comment 3: The authors designed a pH-asymmetric electrolyzer for direct seawater electrolysis. Can the pH maintain on both anode and cathode side during the long operation? The authors should check the pH after long-time cycling. Also, would the local pH near the electrode be different under operating conditions?

Reply: Thanks for the valuable and insightful comments. We collected and tested the pH of anode and cathode electrolytes of the asymmetric electrolyzer over time while running it at 100 mA cm⁻² (**Figure R5a** and new **Figure S27**). As time increased, the pH of cathode electrolyte gradually increases, indicating the consumption of protons for HER, while the pH of anode electrolyte almost keeps constant. After 12 h stability test, the pH of cathode electrolyte increases from 6.9 to 8.8, and the pH of anode electrolyte slightly drops from 14.4 to 14.1, which is consistent with the performance degradation of the asymmetric electrolyzer for direct seawater electrolysis at electrolyte circulation mode. However, we could maintain the pH of cathode electrolyte with a one-way feeding scheme to stabilize the voltage of water electrolysis (reply in comment 5). We also measured the pH of the output cathode electrolyte of the asymmetric electrolyzer over time while running at 100 mA cm⁻² (**Figure R5b** and new **Figure S28**). The pH of the output cathode electrolyte could be maintained at about 8.5 in one-way feeding scheme.

Figure R5. (a) The pH of anode and cathode electrolyte of the asymmetric electrolyzer over time at 100 mA cm^{-2} (electrolyte cycled mode). (b) the pH of cathode electrolyte with a one-way feeding flowing scheme.

Furthermore, we investigated the local pH on the cathode catalyst surface as function of the operating potential. A rotating ring-disk electrode (RRDE) was used to detect the local pH of the electrode surface. The local pH on the ring electrode is calculated from the open circuit potential (OCP) of the ring electrode since the OCP obeys the Nernst equation under the condition that hydrogen oxidation/evolution reactions (HOR/HER) proceed reversibly. Measuring pH on the ring electrode does not disturb the reactions on the disk electrode. In addition, the flow from the disk to the ring electrodes can be described accurately by a convective-diffusion equation, hence the local pH observed at the ring electrode can be converted into the local pH immediately adjacent to the disk electrode (ChemElectroChem 2019, 6, 4750).

To confirm that the OCP of the Pt ring electrode accurately reflected the hydrogen equilibrium potential, the ring OCPs are measured with solutions of various pH values without any reactions at the disk electrode. The time dependence of the ring OCP is shown in **Figure R6a** and new **Figure S23a**. The pH values shown in the figure are those in the bulk solution measured by a pH meter. We take a linear fit of OCPs response to pH of the ring, in which the slope is 58 mV/pH and the intercept is -6 mV , showing a good agreement with Nernst equation (**Figure R6b** and new **Figure S23b**).

Figure R6. (a) Time and (b) pH dependence of open circuit potential (OCP) for ring electrode. The measurement was performed in NaCl solution, and the pH of the bulk electrolyte was changed by adding H₂SO₄ or NaOH.

Next, the measurement of pH on the catalyst surface is performed under different applied potentials. Pt_{SA}-Ni_{6.6}Fe_{0.4}P₃ is scraped off the Ni foam and then loaded on the disk electrode. Linear sweep voltammetry is performed on the catalyst-loaded disk electrode in NaCl solution, and meanwhile, OCPs are recorded on the ring electrode. The pH value of the ring electrode is evaluated from the OCPs using the equation.

$$\text{pH}_{\text{ring}} = \frac{\text{OCP} + 0.006}{-0.058}$$

The pH value of the catalyst-loaded disk electrode can be deduced from the pH value of the ring electrode by the following equation (ChemElectroChem 2019, 6, 4750):

$$C_{\text{H}^+, \text{ring}} - C_{\text{OH}^-, \text{ring}} = N_{\text{D}} \times (C_{\text{H}^+, \text{disk}} - C_{\text{OH}^-, \text{disk}}) + (1 - N_{\text{D}}) \times (C_{\text{H}^+, \text{bulk}} - C_{\text{OH}^-, \text{bulk}})$$

where $C_{\text{H}^+, \text{ring}}$ and $C_{\text{H}^+, \text{disk}}$ are the concentrations of H⁺ on the ring and disk electrodes, respectively, $C_{\text{OH}^-, \text{ring}}$ and $C_{\text{OH}^-, \text{disk}}$ are the concentrations of OH⁻ on the ring and disk electrode, respectively, $C_{\text{H}^+, \text{bulk}}$ and $C_{\text{OH}^-, \text{bulk}}$ are the concentrations of H⁺ and OH⁻ in the bulk solution, respectively, and $N_{\text{D}} = 0.38$ is the collection efficiency of the ring electrode.

The local pH of the catalyst surface against the potential during the HER in NaCl solution is shown in **Figure R7** and new **Figure S24**. When HER occurs, the local pH of the electrode surface will increase, but it is still far lower than the anode pH (14.4), indicating that the chemical potential due to the pH difference can be leveraged to increase the efficiency of the process.

Figure R7. The local pH on cathode electrode surface at different potentials (blue line) and the LSV curves (red line).

The following discussion has been added on the part of “Electrocatalytic performance for seawater electrolysis” and “supplementary information” in the revised manuscript:

“The rotating ring-disk electrode technique was used to quantitatively detect the local pH on the $Pt_{SA}-Ni_{6.6}Fe_{0.4}P_3$ for HER in NaCl solution (Supplementary Note 2 and Supplementary Figs. 23-24)⁴⁴. We find that the pH value on the $Pt_{SA}-Ni_{6.6}Fe_{0.4}P_3$ increases when HER occurs but it is still far lower than the anode pH (14.4), indicating that the chemical potential due to the pH difference can be leveraged to increase the efficiency of the process.” and “The pH of anode and cathode electrolyte of the asymmetric electrolyzer over time at 100 mA cm^{-2} was also collected when the electrolyte was cycled. As shown in Supplementary Fig. 27, the pH of cathode electrolyte increases from 6.9 to 8.8, and the pH of anode electrolyte slightly drops from 14.4 to 14.1, which is consistent with the performance degradation of the asymmetric electrolyzer for water electrolysis if the electrolyte is cycled. However, the pH of the output cathode electrolyte (Supplementary Fig. 28) could be maintained at about 8.5 when operating at 100 mA cm^{-2} in one-way flow feed scheme, and the electrolyte with steady pH can exhibit stable voltage response.”.

Comment 4: The evidence of Pt single atoms is lacking as well. The authors performed Pt L3-edge EXFAS to prove the coordination of Pt-Ni. How would the Pt L3-edge be fitted if assuming the coordination is Pt-Fe? EXAFS fitting can be tricky and the difference between Pt-Ni and Pt-Fe is not obvious. Also, for single atom evidence, HAADF-STEM to see the distribution of Pt over $Ni_{6.6}Fe_{0.4}P_3$ is a more direct evidence.

Reply: We would like to thank the reviewer for the important and valuable suggestion.

In the Pt_{SA}-Ni_{6.6}Fe_{0.4}P₃ catalyst, the atomic ratio of Ni to Fe is as high as 33 to 2. With such a high Ni/Fe ratio as well as the insignificant difference between Pt-Ni and Pt-Fe, we suggest a main Pt-Ni coordination to simplify the modeling of this system in EXAFS fitting. We also fit the Pt L3-edge EXAFS by the Pt-Fe coordination in the same way with Pt-Ni (**Figure R8**).

Figure R8. EXAFS fitting curve of the first-shell of Pt_{SA}-Ni_{6.6}Fe_{0.4}P₃ R-space by Pt-Fe and Pt-O coordination.

FT-EXAFS fitting results of Pt_{SA}-Ni_{6.6}Fe_{0.4}P₃ by Pt-Fe coordination are given in Table. R2. There is no major difference in coordination numbers between two fitting methods. Moreover, the Pt-Fe (2.6 Å, Nat Commun 2022, 13, 6798), Pt-Ni (2.6 Å, Nat Commun 2021, 12, 3783) bonds are difficult to distinguish due to the similar bond length. Therefore, we cannot confirm exactly whether it is Pt-Ni or Pt-Fe coordination. Nevertheless, we chose to simplify model this system using EXAFS fitting only with Pt-Ni coordination due to the relatively high Ni:Fe ratio (33:2). And we have changed the labeling of the text and images from Pt-Ni to Pt-Ni/Fe coordination.

Table R2. FT-EXAFS fitting results of Pt_{SA}-Ni_{6.6}Fe_{0.4}P₃ by Pt-Fe and Pt-O coordination.

	Shell	CN	R (Å)	σ^2	ΔE_0	R factor
Pt _{SA} - Ni _{6.6} Fe _{0.4} P ₃	Pt-O	1.01±0.6	2.02±0.01	0.004	7.73±3.3	0.031
	Pt-Fe	3.21±0.6	2.56±0.01	0.004		

In addition, we supplemented HAADF-STEM to show the distribution of Pt over Ni_{6.6}Fe_{0.4}P₃ for single atom evidence (**Figure R9** and new **Figure S5**). The dispersed bright dots (red circles) in the image confirmed the presence of single-atom Pt in Pt_{SA}-Ni_{6.6}Fe_{0.4}P₃.

Figure R9. The HAADF-STEM images of $\text{Pt}_{\text{SA}}\text{-Ni}_{6.6}\text{Fe}_{0.4}\text{P}_3$.

The following discussion has been added on the section of Electrolyzer design and electrocatalyst characterizations.” in the revised manuscript:

“The high-angle annular darkfield STEM (HAADF-STEM, Supplementary Fig. 5) was used to investigate the distribution of Pt over $\text{Ni}_{6.6}\text{Fe}_{0.4}\text{P}_3$. The bright spots appear on the amorphous structure, corresponding to heavy constituent atoms species, which clearly confirms the immobilization of atomically dispersed Pt atoms”,

“The peak at 2.1 Å for $\text{PtSA-Ni}_{6.6}\text{Fe}_{0.4}\text{P}_3$ is associated with the Pt-Ni/Fe bonds, in which the Pt-Ni and Pt-Fe bonds are difficult to distinguish due to the similar bond lengths^{34,35}”

“A main Pt-Ni coordination is taken to simplify the modeling of this system in EXAFS fitting due to the relatively large Ni:Fe ratio (33:2).”

Comment 5: The stability of the asymmetric electrolyzer at constant current density of 400 mA cm^{-2} , $80 \text{ }^\circ\text{C}$ without iR-compensation in Supplementary Figure 24 decays quick fast. Some explanation is needed, especially using the initial cell voltage for H_2 price calculation is not fair or acceptable at all.

Reply: Thanks the reviewer for the valuable suggestion. The reason for the decay of the stability of the asymmetric electrolyzer at constant current density of 400 mA cm^{-2} , $80 \text{ }^\circ\text{C}$ without iR-compensation in Supplementary Figure 24 is that the electrolyte of cathode and anode is recycled, and the pH of NaCl solution will increase during long-term test, in which the proton will be consumed if the solution is circulated. The reply for comment 3 also explains it. For Supplementary Figure 24, the electricity cost for the asymmetric electrolyzer can be calculated to be \$1.4 per kg of H_2 if the average value of voltage (1.71 V) is used. In addition, we optimize the flow of the cathode

electrolyte with a one-way flow method instead of circulation (**Figure R10** and new **Figure S29**). The asymmetric electrolyzer can keep a stable input voltage of 1.66V at 400 mA cm⁻², 80 °C, corresponding to an electricity cost of 1.36 \$ per kg of H₂. The one-way feed of seawater (Fig. S26) further indicates that keeping the pH stable by one-way flow feed for the cathode electrolyte can stabilize the voltage of the asymmetric system.

Figure R10. The stability test of the asymmetric electrolyzer at constant current density of 400 mA cm⁻², 80 °C with cathode electrolyte one-way feeding scheme.

The following discussion has been added on the section of “Electrocatalytic performance for seawater electrolysis.” in the revised manuscript:

“When using the asymmetric electrolyzer working under industrial operating conditions of 400 mA cm⁻² and 80 °C for water splitting with one-way flow scheme, the operating voltage is only 1.66 V without iR-compensation (Supplementary Fig. 29), which is also significantly lower than the symmetric electrolyzer (1.83 V, Supplementary Fig. 30). The stable voltage proves the stability for the asymmetric electrolyzer with one-way flowing feed scheme to stabilize the pH on the cathode.”

Comment 6: Did the authors consider the electrolyte when calculating the free energy diagram of HER on different catalysts? Solvents and ion solvation play an important role in influencing the energetics.

Reply: Thanks the reviewer for the valuable suggestion. We supplement the calculation of the free energy diagram of HER on catalysts in NaCl solution by an implicit solvation calculation which can consider the solvent effect of the solvents and ion solvation (ChemPhysChem 2017, 18, 2171.). The adsorption sites and reaction paths are the same as the original. After considering the solvent effect and ions, the energy barriers change compared to the original, indicating solvent as well as solvent

ions have an effect on the free energy. However, our conclusion from the DFT calculation remains unchanged. As shown in **Figure R11** and new **Figure 5e**, the rate-determining step is still water dissociation and its energy barrier with Pt_{SA}-Ni_{6.6}Fe_{0.4}P₃ (0.46 eV) is still lower than that of a Pt foil (0.72 eV) and Pt_{SA}-Ni-Fe (0.59 eV), indicating Pt_{SA}-Ni_{6.6}Fe_{0.4}P₃ facilitates the water dissociation of HER in NaCl solution. Besides, the free energy of desorption of H* on Pt_{SA}-Ni_{6.6}Fe_{0.4}P₃ (0.10 eV) is closer to zero compared with that on Pt_{SA}-Ni-Fe (-0.17 eV) and Pt foil (-0.21 eV), indicating a faster desorption of H*. The lower energy barrier of water dissociation and a faster desorption of H* account for the best HER performance of Pt_{SA}-Ni_{6.6}Fe_{0.4}P₃.

Figure R11. Free energy diagram of HER on Pt_{SA}-Ni_{6.6}Fe_{0.4}P₃, Pt_{SA}-Ni-Fe and Pt surfaces by an implicit solvation calculation.

And the details of the calculation were as follows:

Theoretical calculations: The Vienna Ab Initio Package (VASP) was employed to perform all the density functional theory (DFT) calculations within the generalized gradient approximation (GGA) using the Perdew, Burke, and Enzerhof (PBE) formulation (Phys. Rev. B 1996, 54, 11169–11186; Phys. Rev. Lett. 1996, 77, 3865–3868; Phys. Rev. B 1999, 59, 1758-1775). The projected augmented wave (PAW) potentials were applied to describe the ionic cores and take valence electrons into account using a plane wave basis set with a kinetic energy cutoff of 450 eV (Phys. Rev. B 1994, 50, 17953–17979; J. Chem. Phys. 2010, 132, 154104). Partial occupancies of the Kohn–Sham orbitals were allowed using the Gaussian smearing method and a width of 0.05 eV. The electronic energy was considered self-consistent when the energy change was smaller than 10⁻⁶ eV. A geometry optimization was considered convergent when the force change was smaller than 0.03 eV/Å. Grimme’s

DFT-D3 methodology was used to describe the dispersion interactions (J. Chem. Phys. 2000, 113, 9901). For implicit solvation calculations, we used VASPsol (J. Chem. Phys. 2014, 140, 084106), a software package that incorporates solvation into VASP within a self-consistent continuum model. VASPsol, due to its simplicity and low computational costs, has been applied in different electrochemical systems in recent years (ChemPhysChem, 2017, 18, 2171–2190). The energy from DFT is added to the energies from electrostatic interactions between the solute and the solvent and the cavitation energy to create the solute within the solvent. The vacuum spacing perpendicular to the plane of the structure is 20 Å. The Brillouin zone integral utilized the surfaces structures of 2×2×1 monkhorst pack K- point sampling. The free energy was calculated using the equation:

$$G = E_{\text{ads}} + \text{ZPE} - \text{TS}$$

where G, E_{ads} , ZPE and TS are the free energy, total energy from DFT calculations, zero point energy and entropic contributions, respectively.

The following discussion has replaced the original calculation on the part of “Mechanism study” and “DFT calculation details” in the revised manuscript:

“Density functional theory (DFT) calculations were further investigated by an implicit solvation calculation to disclose the enhanced HER kinetics on the amorphous $\text{Ni}_{6.6}\text{Fe}_{0.4}\text{P}_3$ supported Pt single atoms in NaCl solution which can consider the solvent effect of the solvents and ion solvation⁵⁴. The models of $\text{Pt}_{\text{SA}}\text{-Ni}_{6.6}\text{Fe}_{0.4}\text{P}_3$, $\text{Pt}_{\text{SA}}\text{-Ni-Fe}$ and Pt foil were built based on the structural characterization in previous section (Fig. 5d and Supplementary Fig. 39, details in Supplementary Note 5). The energy barrier of HER in NaCl on various catalysts includes two key steps: the water dissociation and H^ desorption (Supplementary Fig. 40 and Note 6)³². As shown in Fig. 5e, the energy barrier of water dissociation, which is the rate-determining step on $\text{Pt}_{\text{SA}}\text{-Ni}_{6.6}\text{Fe}_{0.4}\text{P}_3$ (0.46 eV), is lower than that of Pt foil (0.72 eV). It indicates that $\text{Pt}_{\text{SA}}\text{-Ni}_{6.6}\text{Fe}_{0.4}\text{P}_3$ could facilitate the water dissociation, which is consistent with the in-situ Raman spectroscopy. The $\text{Pt}_{\text{SA}}\text{-Ni-Fe}$ has a higher energy barrier of water dissociation (0.59 eV) compared to $\text{Pt}_{\text{SA}}\text{-Ni}_{6.6}\text{Fe}_{0.4}\text{P}_3$, indicating that the amorphous $\text{Ni}_{6.6}\text{Fe}_{0.4}\text{P}_3$ formed after P-doping is the key structure to promote the water dissociation process. Moreover, the free energy of desorption of H^* on $\text{Pt}_{\text{SA}}\text{-Ni}_{6.6}\text{Fe}_{0.4}\text{P}_3$ (0.10 eV) is closer to zero compared with that on $\text{Pt}_{\text{SA}}\text{-Ni-Fe}$ (-0.17 eV) and Pt foil (-0.21 eV), indicating a faster desorption of H^* . The lower energy barrier of water dissociation and a faster desorption of H^* account for the best HER performance of $\text{Pt}_{\text{SA}}\text{-Ni}_{6.6}\text{Fe}_{0.4}\text{P}_3$.”*

“Theoretical calculations: The Vienna Ab Initio Package (VASP) was employed to perform all the density functional theory (DFT) calculations within the generalized gradient approximation (GGA) using the Perdew, Burke, and Enzerhof (PBE) formulation⁵⁷⁻⁵⁹. The projected augmented wave (PAW) potentials were applied to

describe the ionic cores and take valence electrons into account using a plane wave basis set with a kinetic energy cutoff of 450 eV⁶⁰. Partial occupancies of the Kohn–Sham orbitals were allowed using the Gaussian smearing method and a width of 0.05 eV. The electronic energy was considered self-consistent when the energy change was smaller than 10^{−6} eV. A geometry optimization was considered convergent when the force change was smaller than 0.03 eV/Å. Grimme’s DFT-D3 methodology was used to describe the dispersion interactions⁶¹. For implicit solvation calculations, we used VASPsol⁵⁴, a software package that incorporates solvation into VASP within a self-consistent continuum model. VASPsol, due to its simplicity and low computational costs. The energy from DFT is added to the energies from electrostatic interactions between the solute and the solvent and the cavitation energy to create the solute within the solvent. The vacuum spacing perpendicular to the plane of the structure is 20 Å. The Brillouin zone integral utilized the surfaces structures of 2×2×1 monkhorst pack K- point sampling. The free energy was calculated using the equation:

$$G = E_{\text{ads}} + \text{ZPE} - TS$$

where G , E_{ads} , ZPE and TS are the free energy, total energy from DFT calculations, zero-point energy and entropic contributions, respectively.”.

Reviewer comments, further round review -

Reviewer #1 (Remarks to the Author):

The authors have addressed nearly all the comments from the reviewers. I am pleased to recommend its acceptance in this journal at the present form.

Reviewer #2 (Remarks to the Author):

The authors have fully addressed my questions after the revision. I recommend the acceptance of this work.